# An *Aegilops longissima* NLR protein with integrated CC-BED module mediates resistance to wheat powdery mildew

Chao Ma [1,11], Xiubin Tian[2,3,11], Zhenjie Dong[4], Huanhuan Li[1], Xuexue Chen[2], Wenxuan Liu [1], Guihong Yin[5], Shuyang Ma[2,3], Liwei Zhang[6], Aizhong Cao[4], Cheng Liu[7], Hongfei Yan[8], Sunish K. Sehgal [9], Zhibin Zhang[10], Bao Liu [10], Shiwei Wang [6], Qianwen Liu [1] ✉, Yusheng Zhao [2] ✉ & Yue Zhao [1] ✉

Powdery mildew, caused by *Blumeria graminis* f. sp. *tritici* (*Bgt*), reduces wheat yields and grain quality, thus posing a significant threat to global food security. Wild relatives of wheat serve as valuable resources for resistance to powdery mildew. Here, the powdery mildew resistance gene *Pm6Sl* is cloned from the wild wheat species *Aegilops longissima*. It encodes a nucleotide-binding leucine-rich repeat (NLR) protein featuring a CC-BED module formed by a zinc finger BED (Znf-BED) domain integrated into the coiled-coil (CC) domain. The function of *Pm6Sl* is validated via mutagenesis, gene silencing, and transgenic assays. In addition, we develop a resistant germplasm harbouring *Pm6Sl* in a very small segment with no linkage drag along with the diagnostic gene marker *pm6sl-1* to facilitate *Pm6Sl* deployment in wheat breeding programs. The cloning of *Pm6Sl*, a resistance gene with BED-NLR architecture, will increase our understanding of the molecular mechanisms underlying BED-NLR-mediated resistance to various pathogens.

Wheat (*Triticum aestivum* L., 2*n* = 6*x* = 42, AABBDD) is a major staple crop worldwide, contributing approximately 20% of the total dietary protein and calories for the global human population[1]. Powdery mildew poses a substantial threat to global food security by affecting wheat yield and quality[2]. The identification and deployment of effective resistance genes in wheat varieties is widely recognized as one of the most economical, effective and environmentally sustainable approaches to reduce the impact of wheat powdery mildew. To date, more than 100 alleles of 69 powdery mildew resistance genes (*Pm*)

have been officially designated, of which 19 genes have been cloned[3–6]. These genes include 12 (63.16%) *Pm* genes, namely, *Pm1a*, *Pm2*, *Pm3b/Pm8/Pm17*, *Pm5e*, *Pm12/Pm21*, *Pm41*, *Pm55*, *Pm60* and *Pm69*, which encode nucleotide-binding leucine-rich repeat (NLR) proteins; one gene encoding an ATP-binding cassette (ABC) transporter (*Pm38*); and one gene encoding a hexose transporter (*Pm46*). Recently, several resistance genes containing novel/unusual domains, including *Pm24* encoding a tandem kinase protein (TKP), *Pm4* encoding a putative chimeric protein of a serine/threonine kinase, and *Pm36* encoding a

[1]The State Key Laboratory of Wheat and Maize Crop Science, College of Life Sciences, Henan Agricultural University, Zhengzhou 450046, P. R. China. [2]State Key Laboratory of Plant Cell and Chromosome Engineering, Institute of Genetics and Developmental Biology, Chinese Academy of Sciences, Beijing 100101, China. [3]University of Chinese Academy of Sciences, Beijing 100049, P. R. China. [4]College of Agronomy, Nanjing Agricultural University, Nanjing 210000, P. R. China. [5]The State Key Laboratory of Wheat and Maize Crop Science, College of Agronomy, Henan Agricultural University, Zhengzhou 450046, P. R. China. [6]Department of Plant Pathology, College of Plant Protection, China Agricultural University, Beijing 100083, P. R. China. [7]Crop Research Institute, Shandong Academy of Agricultural Sciences, Jinan 250000, P. R. China. [8]College of Plant Protection, Hebei Agricultural University, Baoding 071001, P. R. China. [9]Department of Agronomy, Horticulture and Plant Science, South Dakota State University, Brookings, SD 57007, USA. [10]Key Laboratory of Molecular Epigenetics of the Ministry of Education (MOE), Northeast Normal University, Changchun 130024, P. R. China. [11]These authors contributed equally: Chao Ma, Xiubin Tian. ✉e-mail: l_qianwen@126.com; yusheng.zhao@genetics.ac.cn; zhaoyue@henau.edu.cn

unique tandem kinase with a transmembrane domain, have been reported[6]. We also reported two powdery mildew resistance genes: *Pm13* encodes a mixed lineage kinase domain-like (MLKL) protein that contains an N-terminal MLKL_NTD domain and a C-terminal STK domain[7], and *Pm57* encodes an unusual TKP protein with putative kinase-pseudokinase domains followed by a von Willebrand factor A (vWA) domain (WTK6b-vWA)[8], which sheds light on various plant resistance strategies. However, more effort is needed to understand the diverse array of complex mechanisms that activate plant resistance.

NLRs constitute the largest class of immune receptors in plants and play a pivotal role in the plant defence surveillance system by monitoring pathogen effectors delivered into the plant cells[9,10]. NLRs share a tripartite domain structure, comprising an N-terminal domain, a central nucleotide-binding (NB) domain, and a C-terminal leucine-rich repeat (LRR) domain. On the basis of their N-terminal domains, NLRs can be further classified into three subclasses: NLRs containing Toll/interleukin-1 receptor (TIR) (TNLs), NLRs with N-terminal CC domains (CNLs), and NLRs with RPW8-like CC (CCR) domains (RNLs)[11]. The N-terminal domains, CC, CCR, and TIR, are expected to initiate downstream signalling, as the expression of these domains alone often triggers immune responses and hypersensitive response (HR)[12]. In addition to these typical tripartite domain NLRs, other NLRs with additional integrated domains (NLR-IDs), including those containing WRKY[13], kinase[14], heavy metal-associated (HMA)[15], and zinc finger (Znf) BED domains[16], exist. NLRs with Znf-BED domains (BED-NLRs) constitute the second largest NLR-ID group after WRKYs[17]. To date, six BED-NLRs linked to resistance to plant pathogens have been isolated. These genes include *Yr5/Yr7/YrSP*, which provide resistance to wheat yellow rust[16]; *Xa1/Xo1*, which confer resistance to rice blight and leaf streak[18,19]; and *Rph15*, which induces resistance to barley leaf rust[20]. The Znf-BED domain, which was originally identified in transposases, possesses DNA binding capabilities[21,22]. It is required for the proper function of BED-NLRs, as demonstrated by mutations in the Znf-BED domain resulting in increased susceptibility[16]. However, the mechanism by which the Znf-BED domain is involved in the pathogen resistance of plant NLRs remains largely unknown. The identification, cloning and characterization of more resistance genes will enhance our understanding of resistance activation and signalling mechanisms, accelerate the deployment of resistance genes and lead to the engineering of new strategies for effective disease management.

Over the past century, breeders have carried out numerous crosses to expand the genetic diversity of wheat via introgression of resistance genes from its wild relatives. More than 200 (42.83%) of the currently designated 467 resistance genes in cultivated bread wheat originated outside the bread wheat gene pool[23]. The introgression of chromosome segments from wild relatives of wheat is a well-established strategy for broadening the genetic diversity of disease resistance genes[24]. *Aegilops longissima* Schw. et Musch. ($2n = 2x = 14$, S$^l$S$^l$) is a diploid S-genome member of the *Sitopsis* section in the Triticeae tribe[25], representing a valuable reservoir of genetic diversity for resistance to stem rusts[26,27], powdery mildew[28,29], Septoria glume blotch (SNB)[30], eyespot[31], and drought stress tolerance[32]. Among these genes, *Pm13* on chromosome 3S$^l$ [25] and *Pm66* on 4S$^l$ [29] were introgressed into wheat. Previously, we identified a powdery mildew resistance gene, designated *Pm6Sl*, on chromosome 6S$^l$#3 from *Ae. longissima*, and transferred it to bread wheat via homoeologous recombination between chromosome 6S$^l$#3 and its wheat counterparts induced by the *ph1b* gene[33,34], which enhances homoeologous recombinants in common wheat. We mapped *Pm6Sl* to the long arm of 6S$^l$#3 in a 42.80 Mb distal interval between the markers *Ael58410* and *Ael57699* via an F$_2$ population generated by crossing the wheat-*Ae. longissima* disomic addition line TA7548 containing *Pm6Sl* with TA3809, a Chinese Spring (CS) *ph1b*-deletion mutant[28].

In this work, we report the cloning of the *Pm6Sl* gene. The function of *Pm6Sl* is validated via ethyl methanesulfonate (EMS) mutagenesis, virus-induced gene silencing (VIGS), and transgenic assays. *Pm6Sl* encodes an NLR with a Znf-BED domain integrated in the coiled-coil (CC) domain and is thus designated the Znf-BED CNL. Additionally, we develop several *Pm6Sl* stocks containing small alien segments along with a diagnostic gene marker for *Pm6Sl*. This study documents a resistance gene for elucidating the molecular mechanisms underlying gene-for-gene interactions between wheat and the *Bgt* pathogen. Moreover, wheat lines with small *Pm6Sl*-harboured segments and a diagnostic marker for rapid deployment of *Pm6Sl* are developed to facilitate powdery mildew resistance breeding in wheat.

## Results

### *Pm6Sl* is finely mapped in a 210 kb interval on the long arm of 6S$^l$#3

To fine map *Pm6Sl*, a secondary *Pm6Sl* segregation population (F$_3$) was produced by self-pollinating previously developed heterozygous *Pm6Sl* recombinant plants in a homozygous *ph1b* background. First, we retrieved the *Ael58410* and *Ael57699* corresponding genomic sequences from the *Ae. longissima* cv. TL05 reference genome (Fig. 1a). We then prioritized the low-copy number regions and developed 20 6S$^l$#3-specific PCR markers (Supplementary Data 1 and 2). Next, we screened 8,000 F$_3$ individuals and identified 105 recombinants lacking either *Ael58410* or *Ael57699*. With the subsequent genotyping of these 105 recombinants via 20 newly developed markers and phenotypic responses to the *Bgt* isolate E26, we narrowed *Pm6Sl* to a 0.93 Mb interval between *Ael03991* (654.11 Mb) and *AelO4335* (655.04 Mb) (Fig. 1b).

Next, we developed five additional markers in the 0.93 Mb region and screened eight 6S$^l$#3 recombinants of two distinct types (U and V) from 800 F$_4$ plants derived from heterozygous *Pm6Sl*-containing recombinants (T27-VI and T3012) (Fig. 1c). Among these, five U-type recombinants displayed both markers *Ael039960* (654.21 Mb) and *Ael16698* (654.28 Mb), whereas three V-type plants (T3012I-26, T3012I-29, T3012II-3) carried only the *Ael16698* marker. However, all eight recombinants were resistant to *Bgt* isolate E26. By integrating the *Bgt* responses of the eight recombinants via five-marker analysis, we further delimited *Pm6Sl* to a 210 kb region (654.21-654.42 Mb) between *Ael39960* and *AelO9126* (Fig. 1c and Supplementary Data 1). Annotation of this 210 kb region revealed seven genes, of which only *TL05.6S01G0714700* (designated *CNL1*) and *TL05.6S01G0715000* (*CNL2*) were annotated to encode disease resistance (R) proteins, whereas the remaining five were predicted to encode unknown functional proteins (Fig. 1d and Supplementary Data 3).

### Mutagenesis and gene silencing indicated *CNL1* to be the best candidate for *Pm6Sl*

The primers *SALF-CNL1* and *SALF-CNL2* (Supplementary Data 2) were designed to amplify *CNL1* and *CNL2* sequences, respectively, from the *Pm6Sl* donor TA7548 via the use of genomic DNA (gDNA) or cDNA. Sanger sequencing of amplicons and comparison of sequences between gDNA and cDNA revealed that the gDNA sequence of *CNL1* from TA7548 is 4496 bp long and contains three exons encoding a protein of 1431 amino acids (Supplementary Data 4). For *CNL2*, the gDNA sequence from TA7548 is 3262 bp long, which transcribes 2979 bp coding sequences (CDSs) in three exons encoding a protein of 992 amino acids (Supplementary Data 5). In addition, both the *CNL1* and *CNL2* sequences were absent in CS (Supplementary Fig. 1). Therefore, the *CNL1* and *CNL2* genes were prioritized as *Pm6Sl* candidate genes.

To determine the causal gene responsible for *Pm6Sl*, a mutational screening protocol was employed to identify potential *Pm6Sl* candidates via 30 independent susceptible mutants generated from EMS-treated TA7548S (Supplementary Fig. 2). Analysis of variation

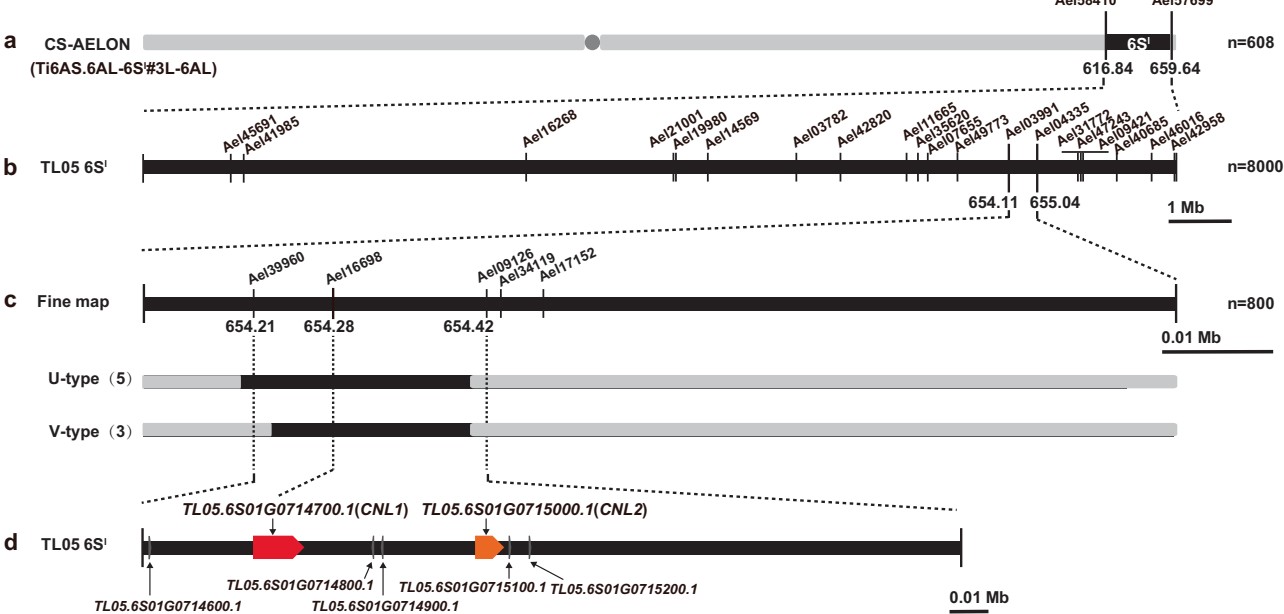

**Fig. 1 | Map-based cloning of *Pm6Sl*. a** Schematic diagram of the wheat-*Ae. longissima* recombined chromosome 6S[l]#3. **b** Resistance assay and marker analysis of 105 6S[l]#3 recombinants narrowed *Pm6Sl* down to a 0.93 Mb interval. **c** Analysis of an additional five markers and evaluation of powdery mildew resistance in two additional distinct types (U and V) of 6S[l]#3 recombinants delimited the *Pm6Sl* interval to 210 kb. The number of recombinants of each type is shown in brackets; all eight recombinants were resistant to the *Bgt* isolate E26. **d** Seven genes discovered within the 210 kb interval, of which only *Ae.longissima.TL05.6S01G0714700* (named *CNL1*) and *Ae.longissima.TL05.6S01G0715000* (named *CNL2*) are annotated to encode resistance proteins.

in *CNL1* in 30 susceptible mutants revealed mutations either introducing a premature stop codon or leading to nonsynonymous amino acid alterations in *CNL1* in 29 mutants, whereas only two mutants (Mut12 and Mut24) presented mutations in *CNL2*, but they also carried nonsynonymous mutations in *CNL1* (Fig. 2a and Supplementary Data 6). Furthermore, we used a barley stripe mosaic virus-induced gene silencing system (BSMV-VIGS) to knock down either *CNL1* or *CNL2* in TA7548 to verify their function. The results revealed that silencing *CNL1* with the BSMV:γ*CNL1* construct resulted in susceptibility of TA7548 plants to powdery mildew (Fig. 2b), whereas no notable phenotypic changes were observed in the plants treated with the BSMV:γ and BSMV:γ*PDS* (phytoene desaturase gene) controls. Conversely, silencing *CNL2* with the BSMV:γ*CNL2* construct did not change resistance levels compared with those in plants treated with the BSMV:γ or BSMV:γ*PDS* controls (Fig. 2b). These results confirmed that *CNL1*, rather than *CNL2*, is the most promising candidate for *Pm6Sl*.

### Transgenic assay verified candidate *CNL1* as *Pm6Sl*

To confirm whether *CNL1* is responsible for powdery mildew resistance in plants containing *Pm6Sl*, the CDS of *CNL1* driven by the maize *Ubi* promoter was introduced into the recipient wheat Fielder, which is susceptible to powdery mildew (Supplementary Fig. 3a). A total of 24 $T_0$ plants were regenerated, of which 20 were *CNL1* positive and 4 were negative. These plants self-pollinated and advanced to generation $T_1$ (Fig. 3a). Among the 20 positive $T_1$ lines, L17 and L21 presented *CNL1* expression levels comparable to those of T3012II-3, a resistant recombinant with a small 6S[l] segment carrying *Pm6Sl*, whereas the other 18 $T_1$ lines presented significantly higher levels of *CNL1* expression (Fig. 3b). An assessment of the resistance of these 20 $T_1$ lines revealed that all the *CNL1*-positive transgenic plants were resistant to the *Bgt* isolate E26, whereas the *CNL1*-negative plants were susceptible (Fig. 3a and Supplementary Fig. 3b). In addition, *CNL1*-positive $T_1$ plants at the heading stage continued to show resistance to powdery mildew, suggesting that *Pm6Sl* is capable of conferring all-stage resistance (Fig. 3c).

*CNL1* function was also validated by a transgenic assay using its native promoter and genomic sequence. The *ProCNL1:CNL1* construct comprising a 9727 bp genomic fragment including a presumed 4569 bp native promoter, 4496 bp exon and intron regions, and 662 bp terminator of *CNL1* was delivered to the Fielder (Supplementary Fig. 4a), generating six independent transgenic $T_0$ seedlings. All plants were positive for *CNL1* and exhibited resistance to powdery mildew akin to the resistance control T3012II-3 (Supplementary Fig. 4b). These results indicate that both normal and elevated *CNL1* expression can effectively confer resistance to *Bgt* isolate E26. Taken together, the results of the mutagenesis, VIGS-induced gene silencing, and transgenic assays consistently confirmed that *CNL1* is the *Pm6Sl* gene.

### *Pm6Sl* encodes an NLR protein containing a CC-BED module

To study the structural characteristics of the Pm6Sl protein, we used the AlphaFold2 artificial intelligence-augmented system to generate a 3D model. The results revealed that *Pm6Sl* encodes an NLR protein that contains a Znf-BED domain in addition to the expected N-terminal CC domain, a central NB-ARC site, and a C-terminal LRR region (Fig. 4a). The CC domain of Pm6Sl comprises four α-helices (α1–4), analogous to that of a conventional CNL protein[35]; nevertheless, the Znf-BED domain is integrated between its α3 and α4 helices to form an α1-3-BED-α4 chimeric structure that we designated the "CC-BED module" (Fig. 4b). Moreover, the incorporation of amino acid changes in susceptible mutants into the Pm6Sl 3D model revealed that all four domains had mutations leading to amino acid changes: two in the CC-BED module, 11 in the NB-ARC domain, and 18 in the LRR domain (Supplementary Fig. 5a). Interestingly, the amino acid changes in the LRR domain in the mutants were mainly localized on the outer α-helices and disordered coils (Supplementary Fig. 5b), rather than on the continuous β-sheet inside the LRR domain, which is generally considered an important region for effector binding[36,37]. Loss of resistance as a result of mutations in all domains of Pm6Sl implies the necessity of these domains in Pm6Sl to mediate resistance to wheat powdery mildew.

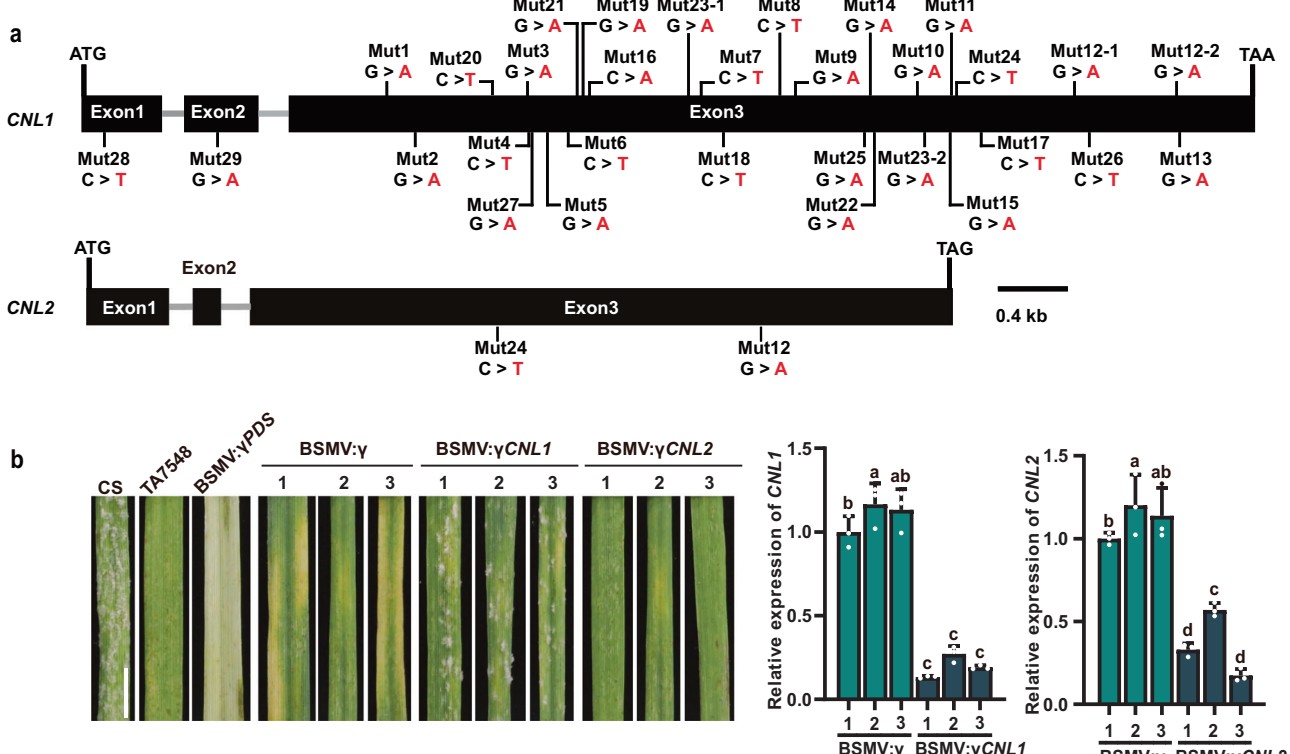

**Fig. 2 | Validation of *Pm6Sl* candidate genes via EMS-induced mutants and VIGS. a** EMS-induced mutation sites in *CNL1* and *CNL2*. Two mutant lines, Mut12 and Mut23, carry two mutations in *CNL1*, and two mutant lines, Mut12 and Mut24, carry nonsynonymous mutations in *CNL1* and *CNL2*. **b** BSMV-VIGS-mediated functional validation of *CNL1* and *CNL2*. Representative leaves showing the resistance phenotype of TA7548 plants subjected to the silencing of *CNL1* (BSMV:*γCNL1*) or *CNL2* (BSMV:*γCNL2*) through BSMV-VIGS; BSMV:*γPDS* was used as a control to illustrate the effect of gene silencing, and BSMV:*γ* was used as an empty vector control. Images show a representative phenotype of the third leaf at 7 days post inoculation (dpi) from seedlings infected with *Bgt* isolate E26. Scale bars = 0.5 cm.

The relative expression of *CNL1* and *CNL2* in the gene-silenced plants was analysed for silencing efficiency via quantitative real-time PCR (qRT–PCR). The *TaACTIN* gene was used for expression normalization. The infected leaves of each plant were mixed as one sample. qRT–PCR was performed three times for each sample, and each time as a technical replicate. The values are the means of three technical replicates with standard deviation (SD) as error bars. Different lowercase letters above the bars denote significant differences at the $p < 0.05$ level (one-way ANOVA). *CNL1* and *CNL2* were efficiently silenced in infected leaves of TA7548 individuals (presented by no. 1, 2, 3), while they were normally expressed in BSMV:*γ* infected leaves. Source data are provided as a Source Data file.

## Pm6Sl is involved in cell death

To explore how *Pm6Sl* provides resistance to the pathogen, T3012II-3 and CS seedlings were infected with *Bgt* isolate E26 and stained with trypan blue (TPN). Compared with that in CS seedlings, increased cell death, as shown by deeper staining, was observed in T3012II-3 seedlings, with a proportion of 38.3% cell death at 48 hpi, which was 3.4 times greater than that in CS seedlings (11.3%) (Fig. 4c). Moreover, a diaminobenzidine (DAB) staining assay revealed that the T3012II-3-infected cells accumulated significantly higher and more stable levels of intracellular reactive oxygen species (intraROS) than the CS-infected cells did (Fig. 4d and Supplementary Fig. 6). These results suggested that *Pm6Sl* may mediate resistance to wheat powdery mildew by promoting cell death.

Previous studies have shown that host cell death in response to effector-triggered immunity is controlled by the CC domain of NLR proteins[38]. To elucidate the mechanism by which *Pm6Sl* triggers cell death, a transient assay with *Nicotiana benthamiana* leaves was performed. The results revealed that only the CC-BED module triggered cell death, whereas neither the full-length Pm6Sl protein nor the other truncated Pm6Sl N-terminal proteins, including α1-3, α1-3-BED, α1-4 (CC) protein or Znf-BED, exhibited this ability (Supplementary Figs. 7 and 5a). In addition, two mutated CC-BED modules, CC-BED[L21F] and CC-BED[M172I], which convert TA7548S from resistant to susceptible to powdery mildew, retained the ability to induce cell death in *N. benthamiana* (Fig. 5b). These findings suggested that the integrity of the

CC-BED module is crucial for inducing cell death, at least under the tested conditions.

Structural prediction analysis was also conducted to determine the presence of the N-terminal CC-BED module in the six other identified resistant BED-NLRs, including *Yr5/Yr7/YrSP* from wheat, *Xa1/Xo1* from rice, and *Rph15* from barley. The results revealed that all the BED-NLR proteins also had a CC-BED module at their N-terminus, which was comparable to that of Pm6Sl (Supplementary Fig. 8). Nevertheless, transient expression of CC-BED, CC (α1–4), and Znf-BED in *N. benthamiana* leaf cells revealed that, unlike Pm6Sl, none of the truncated proteins of these six BED-NLRs induced cell death under the tested conditions (Fig. 5c and Supplementary Fig. 9). Therefore, it is proposed that the CC-BED module represents a conserved structure for BED-NLRs, however, the module may function differently in BED-NLR-mediated resistance.

## Pm6Sl relationships with NLRs from Gramineae species

To investigate the evolutionary origin of Pm6Sl, we first searched for Pm6Sl homologues in the National Center for Biotechnology Information (NCBI) nonredundant protein sequence database. The results revealed that the sequences of the obtained proteins presented low similarity (<75% identity) to those of Pm6Sl (Supplementary Data 7). A subsequent comparative analysis of published genome sequences revealed that Pm6Sl orthologues were absent on homoeologous Group 6 chromosomes of species related to wheat (Supplementary

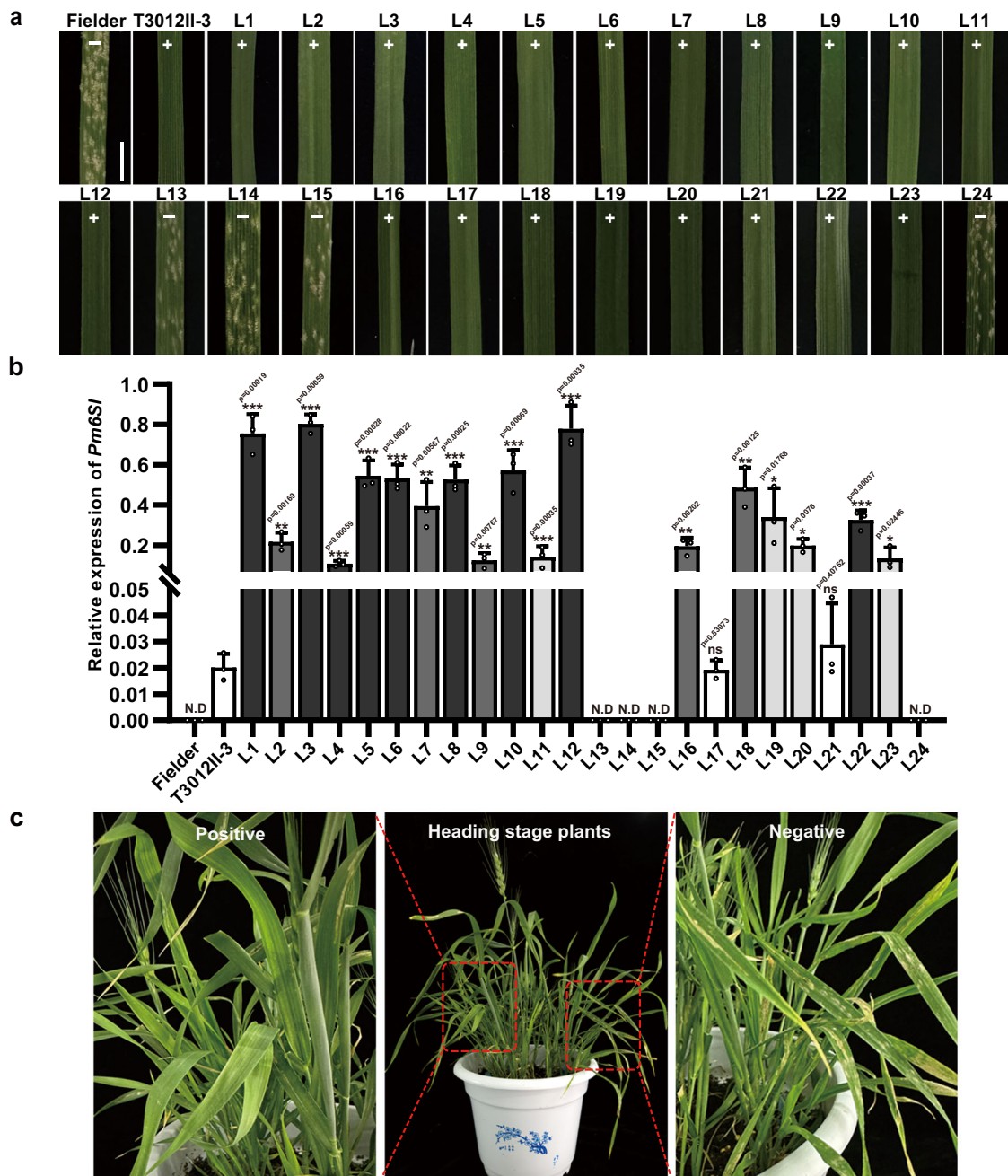

**Fig. 3 | Transgenic assay of the *Pm6Sl* candidate gene *CNL1*. a** Resistance assay of transgenic T1 lines 10 dpi with the *Bgt* isolate E26, with the receptor Fielder as a susceptible control and recombinant T3012II-3 with a small 6Sl segment carrying *Pm6Sl* as a resistance control. "+" indicates the presence of *CNL1*, whereas "–" indicates the absence of *CNL1*. A representative leaf segment phenotype from one plant in each T1 line is shown. Scale bar, 0.5 cm. **b** qRT–PCR analysis of *CNL1* expression in positive T1 plants and T3012II-3 plants at the seedling stage. Data are mean ± SD from three biological replicates. N.D: not detected. Asterisks (*, **, ***) represent significant differences at the $p < 0.05$, $p < 0.01$, and $p < 0.001$ levels, respectively (two-tailed Student's *t* test). Three biological replicates, two leaves from two resistant seedlings per replicate, were carried out for each T1 line. **c** T1 transgenic plant response to *Bgt* isolate E26 at the heading stage. Positive plants harbouring *CNL1* are resistant to powdery mildew, whereas negative control plants in the same pot are susceptible. Source data are provided as a Source Data file.

Fig. 10). Therefore, it is assumed that Pm6Sl may be present exclusively in *Ae. longissima*, although further research is necessary to substantiate this assumption.

To explore the relationships of Pm6Sl with other known NLR proteins, phylogenetic analysis was performed using Pm6Sl and six other cloned BED-NLR proteins, along with 174 known NLR proteins in the Gramineae family. The results indicated that Pm6Sl was most closely related to Rph15 from barley (Supplementary Fig. 11). However, sequence alignment analysis revealed very low amino acid sequence identity (48.8%) between Rph15 and Pm6Sl

(Supplementary Fig. 12). Furthermore, phylogenetic tree analysis of the Znf-BED domains revealed that Pm6Sl is distinct from the Znf-BED domains of the other six cloned BED-NLR proteins, which are highly conserved among proteins annotated in homoeologous group 6 chromosomes of wheat-related species (Supplementary Fig. 13). In contrast, there was 38–45% identity between the ZnF-BED domain of Pm6Sl and those of the six resistant BED-NLR proteins (Supplementary Fig. 14). Therefore, Znf-BED domain integration in Pm6Sl is likely independent of those in the six characterized BED-NLR proteins.

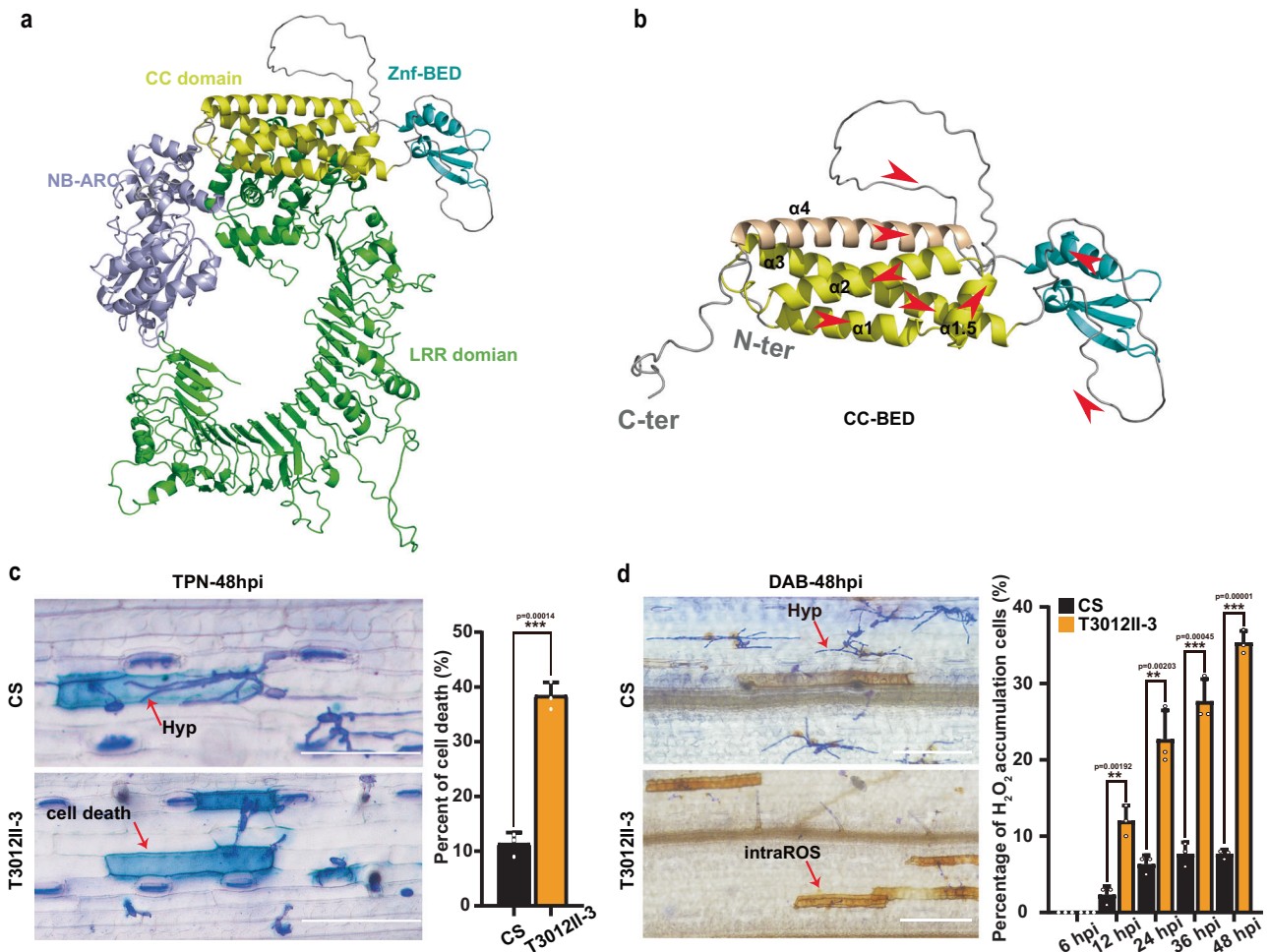

**Fig. 4 | Protein structure prediction and host response of Pm6Sl. a** 3D model of the Pm6Sl protein predicted via AlphaFold2. **b** 3D model of the Pm6Sl CC-BED module predicted via AlphaFold2. **c** TPN-Coomassie blue-stained leaf sections and cell death statistics for infected leaves of T3012II-3 and CS at 48 hpi with *Bgt* isolate E26. Scale bar, 100 μm. **d** Leaf segments with DAB-Coomassie blue and $H_2O_2$ accumulation cell statistics at 48 hpi with *Bgt* isolate E26. Brown staining indicates intracellular reactive oxygen species (IntraROS), while blue staining indicates *Bgt* fungus. Scale bar, 100 μm. Hyp: hyphae. The values are the means with SD as error bars. **p < 0.01, ***p < 0.001 (two-tailed Student's *t* test). Source data are provided as a Source Data file.

## Development of *Pm6Sl* germplasms and a diagnostic gene marker

During *Pm6Sl* cloning, multiple recombinant stocks of CS-*Ae. longissima* containing *Pm6Sl* were developed, with T3012II-3 encapsulating a 0.21 Mb fragment carrying *Pm6Sl* (Supplementary Data 1), equivalent to approximately 0.03% of chromosome 6S$^l$ (666.21 Mb) of *Ae. longissima* cv. TL05. Agronomic traits, such as plant height, number of tillers per plant, spike length, number of spikelets per spike, number of grains per spike, grain length, grain width, and 1000-grain weight, of T3012II-3 were not significantly different from those of CS (the introgression parent), suggesting the absence of deleterious linkage drag associated with *Pm6Sl* in T3012II-3 plants (Fig. 6a, b). Simultaneously, a gene marker, *pm6sl-1*, specific to *Pm6Sl*, was designed (Fig. 6c and Supplementary Data 2). A total of 112 wheat cultivars or advanced breeding lines were evaluated using this marker and *Pm6Sl* stock lines as controls and displayed no *Pm6Sl* loci (Supplementary Data 8), indicating its breeding utility as a diagnostic gene marker for *Pm6Sl*.

To further investigate the value of *Pm6Sl* in wheat breeding, we performed a resistance spectrum assay using 36 *Bgt* isolates with different virulence spectra collected from primary wheat-growing provinces of China. Our results revealed that T3012II-3, harbouring *Pm6Sl*, exhibited immunity or resistance (IT 0–2) against 35 of these *Bgt* isolates at the seedling stage (Supplementary Data 9). These findings

confirmed that *Pm6Sl* confers resistance to multiple *Bgt* isolates. In addition, as previously mentioned, *Pm6Sl* is capable of conferring all-stage resistance, thus reinforcing its significant potential for application in wheat breeding.

## Discussion

The wild relatives of wheat harbour a diverse array of resistance genes, many of which have been introgressed into common wheat through the development of wild wheat species amphidiploids, addition or substitution lines, and translocation lines[39]. However, the process of cloning these alien genes in common wheat via map-based cloning was previously challenging because of the suppression of homologous recombination between wheat and wild species, the limited number of chromosome-specific markers, and the complexity of identifying candidate genes[40]. In recent years, the exploitation of *ph1b* deletion mutants[33] has enabled increased homologous recombination among wild wheat species, thus accelerating map-based cloning of genes from wild wheat relatives. Further integration of mutagenesis and next-generation sequences has resulted in map-based cloning of many resistance genes, such as *Pm21*[41,42], *Fhb7*[43], and *Lr47*[44]. Mapping and cloning alien genes from wild relatives in a hexaploid wheat background offers an additional advantage in the form of ready-to-deploy germplasm that contains shortened segments harbouring resistance

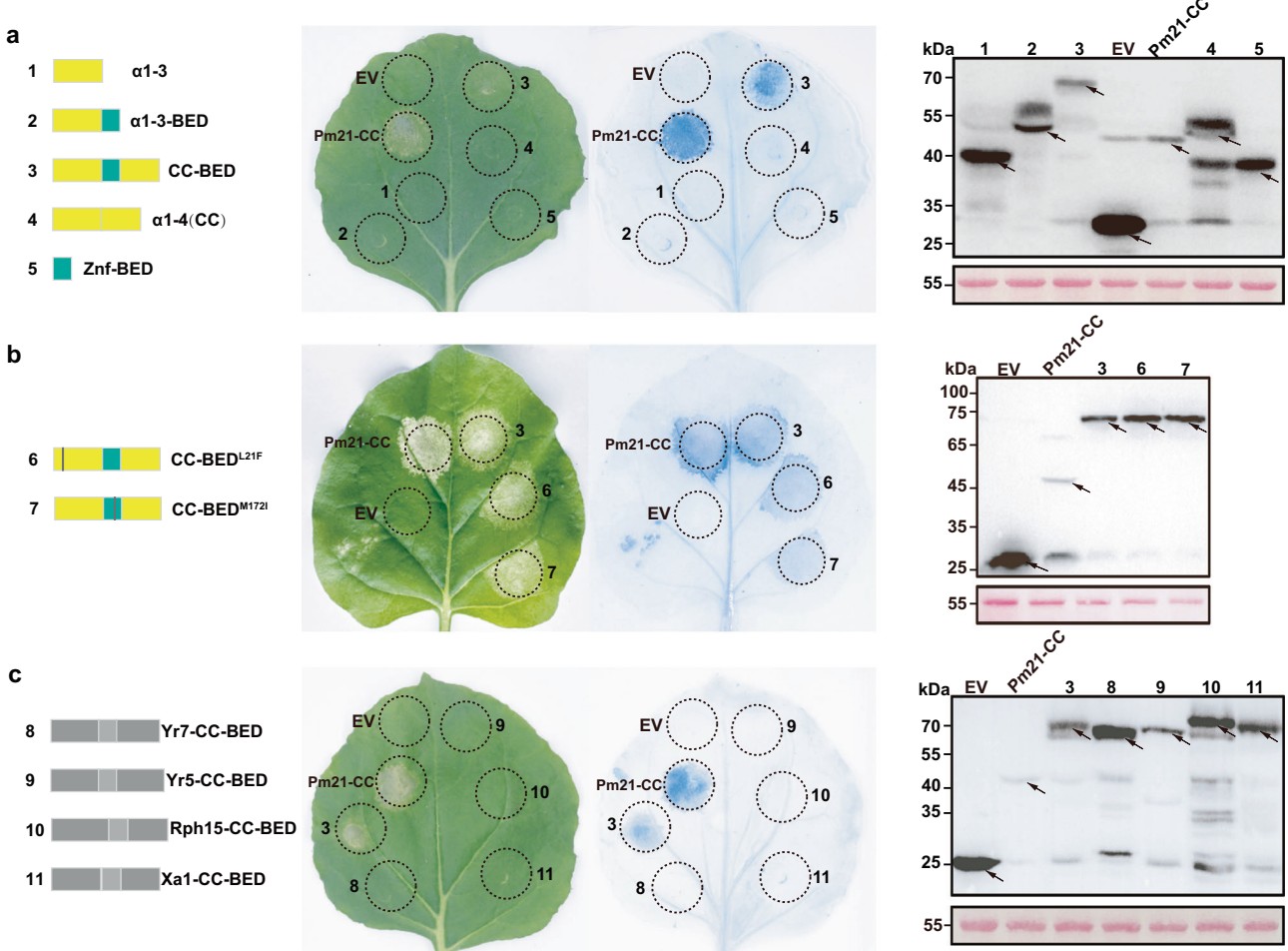

**Fig. 5 | Lethal ability analysis of different BED-NLR CC-BED modules in *N. benthamiana* leaves. a** Transient expression, TPN staining, and western blot of different truncated forms of the Pm6Sl CC-BED module in *N. benthamiana* cells. 1–5: schematic diagram illustrating the truncated forms of the Pm6Sl CC-BED module; Empty vector (EV), negative control; Pm21-CC, positive control. **b** Transient expression, TPN staining, and western blot of the Pm6Sl CC-BED module and two mutants (CC-BED^L21F and CC-BED^M172I) identified in the Pm6Sl CC-BED module through EMS mutagenesis in *N. benthamiana* cells. 6-7: Schematic diagram depicting two mutants (L21F and M172I) identified within the Pm6Sl CC-BED module via EMS mutagenesis. **c** Transient expression and TPN staining of the CC-BED proteins Pm6Sl, Yr7, Yr5, Rph15 and Xa1 in *N. benthamiana* cells. 8–11: Schematic diagram of the CC-BED domains of Yr7, Yr5, Rph15 and Xa1. The total proteins were extracted from the leaves of *N. benthamiana* at 48 hours after infection, and Ponceau S was employed as a stain for normalizing the total protein content. Following this, equivalent samples were utilized, and the tagged proteins were detected through western blot analysis using an anti-GFP antibody. The black arrows indicate the location of the target protein. The experiment was independently repeated three times with similar results. Source data are provided as a Source Data file.

genes from wild relatives suitable for wheat breeding. In this study, we successfully isolated *Pm6Sl* from *Ae. longissima* in common wheat by combining map-based cloning with *ph1b*-induced homologous recombination and compared the candidate gene sequences of loss-of-function mutants and *Pm6Sl* donors. Furthermore, we developed several wheat-*Ae. longissima* recombinant stocks containing small *Pm6Sl* segments without significant adverse agronomic effects. These outcomes highlight the benefits of our map-based cloning strategy and the development of germplasms with shortened alien segments carrying alien genes that increase wheat resistance.

To date, 19 wheat *Pm* genes have been cloned, and their encoding proteins can be divided into four classes: typical NLR proteins, kinase fusion proteins (KFPs), ABC transporters, and hexose transporters[3–6]. Although 12 of the 19 cloned *Pm* genes encoded NLR proteins, none of the cloned *Pm* genes prior to *Pm6Sl* encoded an NLR protein with an additional integrated domain. In this study, *Pm6Sl* from *Ae. longissima* was shown to encode an NLR containing a Znf-BED domain. The cloning of *Pm6Sl* provides a valuable resistance gene for revealing the molecular mechanism of resistance to

powdery mildew in wheat. To date, in addition to Pm6Sl, only six other NLR-IDs possessing a Znf-BED domain conferring resistance to barley or wheat rust, rice blight and leaf streak have been cloned from different crop species. Furthermore, two Znf-BED-containing non-NLRs, YrNAM, which confers resistance to wheat stripe rust, and the NAC transcription factor protein Rph7, which regulates resistance to barley leaf rust, were recently cloned[45,46]. Pm6Sl is a cloned BED-NLR architecture used to control powdery mildew resistance in wheat. In addition, by 3D structure prediction of Pm6Sl and the other six cloned pathogen-resistant BED-NLR architectures via Alphofold2, we discovered that all these genes had an N-terminal CC domain composed of four helixes (α1-4), and the Znf-BED domain was integrated between α3 and α4 of the CC domain, forming a CC-BED module (Supplementary Fig. 8). In this study, two mutations in the CC-BED domain of Pm6Sl, which convert leucine to phenylalanine and methionine to isoleucine, respectively, were identified (Supplementary Fig. 5a). These two mutations resulted in susceptibility to both Mut28 and Mut29, indicating that CC-BED is necessary for Pm6Sl to mediate resistance.

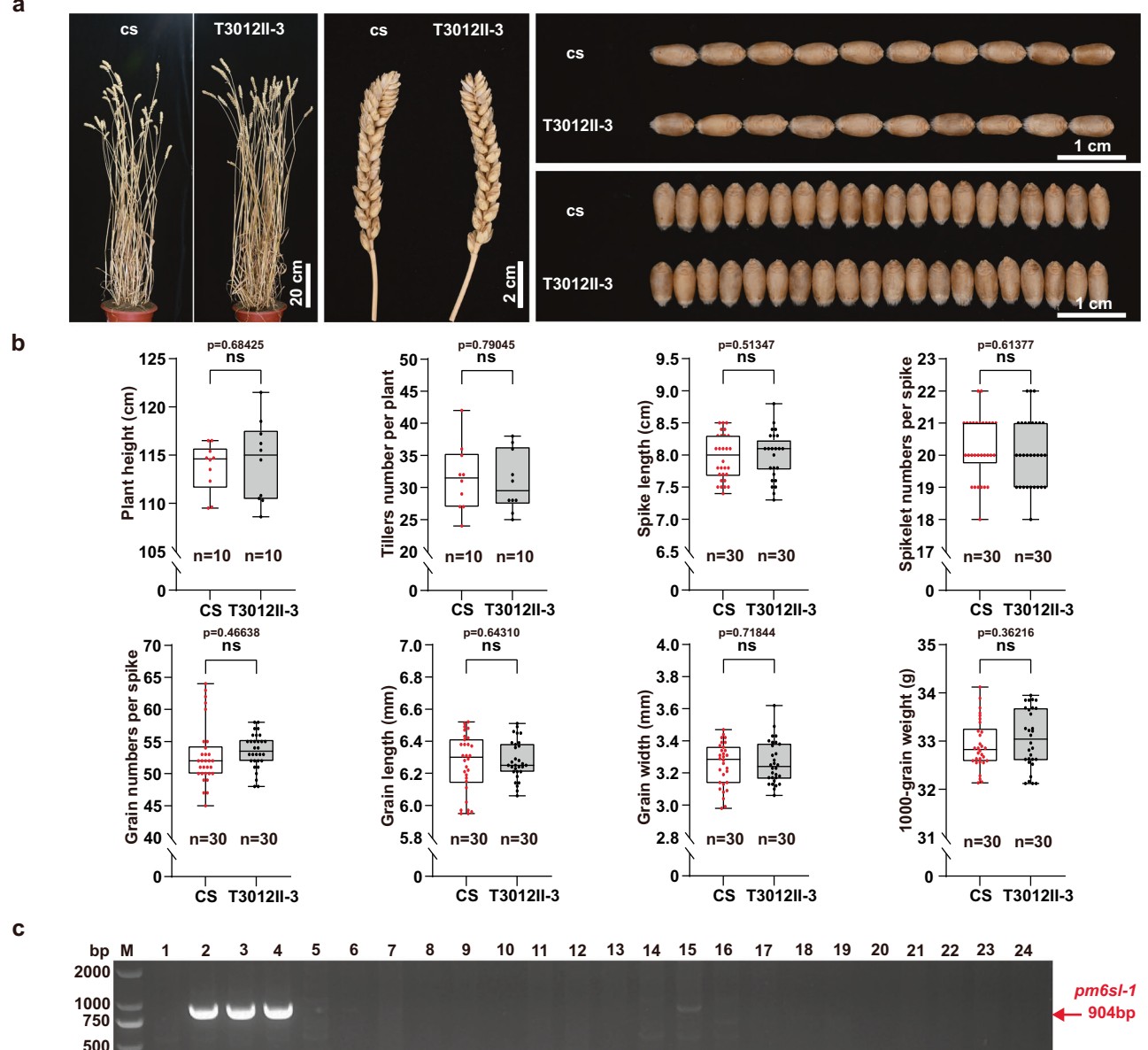

**Fig. 6 | Investigation of the agronomic traits of T3012II-3 and development of the gene marker *pm6sl-1*. a** Visual phenotypes of adult plants, spikes and grains of T3012II-3 (*Pm6Sl +ve*) and CS (*Pm6Sl −ve*) plants. **b** Comparing the agronomic traits between T3012II-3 and CS in plant height (*n* = 10), number of tillers per plant (*n* = 10), spike length (*n* = 30), number of spikelets per spike (*n* = 30), number of grains per spike (*n* = 30), grain length (*n* = 30), grain width (*n* = 30), and 1000-grain weight (*n* = 30). The data are displayed as box and whisker plots with individual data points. The error bars represent the maximum and minimum values. Center-line, median; box limits, 25th and 75th percentiles. Significant differences were determined by two-tailed Student's *t* test. ns: no significant difference. **c** Electrophoresis patterns of the *Pm6Sl* gene marker *pm6sl-1*. M: 2 kb DNA Ladder Marker; 1: CS; 2-4: *Pm6Sl* stocks TA7548, TA7548S, and T3012II-3; 5–24: common wheat varieties Aikang58, Guomai301, Ningchun4, Zhoumai22, Kenong9204, Nongda399, Jimai22, Fielder, Xiaoyan81, Xiaoyan60, Yannong19, Jing411, Xinong528, Fengchan3, Aiqianniu, Nonglin11, Pingan602, Pingan0518, Pastor and Japper. The PCR amplicons of *Pm6Sl* are indicated by red arrows. The experiment was independently repeated three times with similar results. Source data are provided as a Source Data file.

Zinc finger proteins constitute a superfamily of proteins that play crucial roles in diverse aspects of plant functions. Recent reports highlight the importance of zinc finger domains in host–pathogen interactions. For example, TaZF, a wheat zinc finger protein, facilitates Pm2a-mediated recognition of AvrPm2, and in particular, TaZF's zinc finger domain alone is sufficient for interaction with AvrPm2[47]. In addition, Xa1, a rice BED-NLR protein, recognizes the transcription activator-like (TAL) effectors of *Xanthomonas oryzae* pv *oryzae* (*Xoo*), AvrXa7 and Xoo1132, through its BED domain, which forms a complex with these two TAL effectors. Furthermore, the AP2/ERF-type transcription factor OsERF101 may regulate recognition and thus positively regulate Xa1-mediated immunity[48]. Even so, the BED domain was unlikely the sole domain of a BED-NLR protein determining its recognition specificity to pathogen effectors, as ZnF-BEDs with identical BED domain sequences, such as Yr5 and YrSP[16], Xa1 and Xo1[18], can confer resistance to different isolates of a pathogen or different pathogens. In this study, although the expression of the Pm6Sl CC-BED domain in *N. benthamiana* triggered a cell death response, the individual CC or BED domains, as well as other N-terminal truncations, failed to elicit this response (Fig. 5a). This further corroborates the hypothesis that the CC and BED domains of a BED-NLR function as an integrated CC-BED module. Additionally, the two mutant CC-BED types (CC-BED^L21F and CC-BED^M172I) maintained an *N. benthamiana* cell death response (Fig. 5b). These findings suggested that these two mutations resulted

in the loss of Pm6Sl resistance, most likely by influencing other processes, such as pathogen recognition, rather than directly affecting the induction of cell death. However, reports on the roles of Znf-BEDs in plant immunity are scarce, necessitating further experimental evidence to elucidate their molecular mechanism.

Furthermore, on the basis of transient expression in *N. benthamiana* leaves, we observed that the CC-BED modules are structurally conserved in crop-derived BED-NLRs that confer resistance to various pathogens, but the functions of the module most likely vary in different proteins; for example, the CC-BED module of Pm6Sl, unlike the other six BED-NLRs, triggered cell death. In addition, as revealed by sequence alignment and phylogenetic analysis, the sequences of both the Znf-BED domain and CC-BED module of Pm6Sl are distinct from those of six other cloned BED-NLRs (Supplementary Fig. 15) and may result in functional diversification of CC-BED modules. Nevertheless, the roles of CC-BED modules in BED-NLR-mediated resistance to diverse pathogens have yet to be experimentally verified. Pm6Sl provides a new member of the BED-NLR family for understanding the molecular mechanisms of Znf-BED-containing architecture-mediated resistance to diverse pathogens.

In conclusion, we isolated *Pm6Sl* from *Ae. longissima*, which was integrated into a common wheat background, via positional cloning with *ph1b*-induced homoeologous recombination, integrating gene function validation via mutagenesis, gene silencing and transgenic assays. *Pm6Sl* encodes an NLR architecture with a CC-BED module, which is structurally conserved for crop BED-NLRs with resistance to diverse pathogens, enhancing our understanding of the NLR protein-mediated resistance mechanism. Furthermore, multiple wheat-*Ae. longissima* recombinant stocks encapsulating short *Pm6Sl* fragments devoid of detrimental agronomic traits and a diagnostic gene marker, *pm6sl-1*, were generated to increase the deployment of *Pm6Sl* in wheat breeding.

## Methods
### Plant materials
The plant materials used in this study included common wheat CS (TA3808) and the CS *ph1b* mutant (TA3809), which can induce high-frequency homologous pairing and recombination; CS-*Ae. longissima* disomic 6S[l]#3 addition line TA7548, where a pair of 6S[l]#3 chromosomes carrying *Pm6Sl* from *Ae. longissima* was added to the CS background; CS-*Ae. longissima* 6S[l]#3[6 A] substitution line TA7548S, where a pair of CS 6 A chromosomes were replaced by 6S[l]#3 pairs; CS-*Ae. longissima* 6S[l]#3 recombinants T24, T27 and R43, containing *Pm6Sl*-harbouring segments; and Fielder, a common wheat variety used as a receptor for wheat genetic transformation (Supplementary Data 8). Among these materials, TA3808, TA3809, TA7548, and TA7548S were obtained from Wheat Genetics Resource Center (WGRC), Kansas State University, USA, and Fielder was obtained from the Shandong Academy of Agricultural Sciences, China. The CS-*Ae. longissima* 6S[l]#3 recombinants were developed from a homozygous *ph1b* population segregating for 6S[l]#3 that was derived from self-pollinated hybrids of TA7548 crossed with TA3809. In addition, 112 common wheat varieties or advanced lines collected from wheat breeders were used to validate the diagnostic value of gene markers for *Pm6Sl* selection (Supplementary Data 8). *N. benthamiana* was also used as a transformation receptor for transient expression analysis. All the plant materials were maintained at the Experimental Station of Henan Agricultural University, Zhengzhou, China.

### Pathogen inoculation and phenotype identification
*Bgt* isolate E26 and 35 other genetically divergent isolates collected from various regions of China (Supplementary Data 9) were used to evaluate the powdery mildew resistance spectrum of *Pm6Sl*, while the isolate E26 was used for all other resistance assays in this study. When the first leaves were fully expanded, the seedlings were inoculated with *Bgt* isolates and grown in an incubator maintained at 20–22 °C, 16 h light/8 h dark cycle and approximately 95% relative humidity[24]. At 7–10 post inoculation (dpi), the plant responses to *Bgt* were scored on a 0 to 4 scale (0, 1, 2, 3, and 4), with ITs 0, 1 and 2 indicating resistance and ITs 3 and 4 indicating susceptibility.

### Mutagenesis
In total, 6000 seeds of the CS-*Ae. longissima* disomic 6S[l]#3[6 A] substitution line TA7548S were subjected to EMS treatment to generate loss-of-function mutants. The seeds were exposed to 0.6% (v/v) EMS solution and shaken at 150 rpm for 16 h in darkness at room temperature. Approximately 80 EMS-treated seeds were sown in rows of 2 × 0.3 m in the field. A total of 906 $M_1$ plants were harvested. Thirty $M_2$ seeds from each $M_1$ plant, along with CS (susceptible control) and TA7548S (resistant control) plants, were germinated and cultivated in an incubator at 22–25 °C with 60% humidity and a 16-h light/8-h dark photoperiod. Once the leaves were fully expanded, the seedlings were inoculated with *Bgt* isolate E26 and grown at 20–22 °C with 95% humidity and 16 h of supplemental light. After 10 dpi, 41 $M_2$ individuals with ITs of 3–4 from 30 $M_2$ lines segregating for powdery mildew resistance were selected for further validation via the 6S[l]#3-specific markers *Ael69501*, *Ael67319*, *Ael63185*, and *Ael42958* to exclude susceptible plants caused by 6S[l]#3 loss or seed impurity. Three $M_3$ seedlings from one of the 41 $M_2$ susceptible plants for each of the three replicates were subsequently reassessed for susceptibility. The gDNA and cDNA of thirty independently susceptible $M_3$ mutant lines derived from different $M_1$ plants were extracted and used to amplify candidate gene sequences to identify mutation sites through Sanger sequencing.

### RNA extraction and qRT–PCR analysis
Fresh leaves obtained from plant material were cryopreserved in liquid nitrogen and used for RNA extraction. TRIzol reagent (Cat. No. R401, Vazyme, Nanjing, China) was used to extract total RNA samples, and 1 μg of total RNA was then subjected to reverse transcription via HiScript III RT SuperMix for qPCR (+ gDNA wiper) (Cat. No. R323, Vazyme, Nanjing, China) following the manufacturer's instructions. The resulting reverse transcripts were diluted fivefold and used for quantitative real-time RT–PCR (qRT–PCR). A 10 μl qRT–PCR mixture comprising 5 μl of Taq Pro Universal SYBR qPCR Master Mix (Cat. No. Q712, Vazyme, Nanjing, China), 0.2 μl of 10 mM forward and reverse primers, and 4.6 μl of diluted cDNA was used. The PCR amplification started at 95 °C for 30 s, followed by 40 cycles of 95 °C for 10 s and 60 °C for 30 s. The instrument's default melting curve acquisition procedure was then utilized. Wheat *TaACTIN* (*TraesCS5B02G124100*) was used as an internal reference gene[49]. qRT–PCR was performed via an ABI StepOne plus fluorescence quantitative PCR instrument (ABI, USA), and transcription levels were calculated via the comparative CT method[50]. The primers used to determine *Pm6Sl* transcript levels are listed in Supplementary Data 2.

### Fine mapping of gene *Pm6Sl*
Molecular markers specific to the 6S[l]#3 chromosome were designed based on *Ae. longissima* cv. TL05 genomic reference. Genomic DNA was obtained via the CTAB method. The 15 μL PCR mixture contained 2.0 μL of template gDNA (100 ng/μL), 1.0 μL of each primer (5.0 μmol/L), 7.5 μL of Taq MasterMix (Cat. No. CW0862; Cowin Biotech Co. Ltd., Beijing, China) and 3.5 μL of ddH₂O. PCR amplification was performed using a Touchdown-63 reaction procedure[51]. Following electrophoresis on a 2.0% agarose gel with 4SGelred nucleic acid dye (Cat. No. A616697, BBI Life Science Corporation, Shanghai, China), the PCR products were visualized using GBOXSYDR4/1494 Syngene Automated Gel Imager (Syngene, Frederick, MD21704, USA) under UV light.

Heterozygous CS-*Ae. longissima* 6S[l]#3 recombinants T24, T27 and R43, in a homozygous *ph1b* background, were used to generate secondary populations segregating for *Pm6Sl*. Two rounds of gene mapping were conducted. In the first round, fifty seedlings at a fully

extended leaf stage were inoculated with *Bgt* isolate E26, and resistance was assayed for each of the 6S$^l$#3 recombinants. The secondary population consisting of seedlings that segregated for powdery mildew resistance was used to generate 6S$^l$#3 recombinants via the markers *Ael5841O* and *Ael57699*, which flank the *Pm6Sl*-containing interval. Plants lacking either marker were considered 6S$^l$#3 recombinants and were further analysed via the 20 markers in the candidate gene region (Supplementary Data 2). Fifteen unfolded first-leaf seedlings from each F$_2$ progeny of the 6S$^l$#3 recombinants were inoculated with *Bgt* E26 to evaluate the resistance of the recombinants. The recombinant was determined to have a susceptible phenotype if all 15 F$_2$ seedlings were susceptible. On the other hand, if the 15 F$_2$ seedlings segregated for resistance and only resistant seedlings presented with the 6S$^l$#3-specific marker, the recombinant parent was confirmed to be resistant. *Pm6Sl* mapping was performed carefully by integrating genotypic and phenotypic data from each 6S$^l$#3 recombinant between markers that were present in all resistant recombinants but absent in susceptible recombinants. In the second round, the resistant recombinants T27-6 and T3012 (Supplementary Data 1), which were selected in the first round, were used to generate a secondary population segregating for *Pm6Sl*. 6S$^l$#3 recombinants were identified via the markers *Ael0433S* and *Ael03991* flanking *Pm6Sl* identified via first-round mapping. The genotype of each recombinant was determined via five newly designed markers, *Ael3996O*, *Ael16698*, *Ael09126*, *Ael34119*, and *Ael17152*, within the *Pm6Sl* mapping interval (Supplementary Data 1 and 2). The responses of each recombinant to *Bgt* isolate E26 were evaluated to further finely map *Pm6Sl* via the method described for initial mapping.

### Identification and cloning of the candidate genes

*Pm6Sl* candidate genes were identified by aligning the sequences of *Pm6Sl* flanking markers with the genomic reference sequences of *Ae. longissima* cv. TL05. *CNL1* and *CNL2*, two annotated resistance (R) genes in the *Pm6Sl* interval, were selected as *Pm6Sl* candidates. PCR primer pairs were designed to isolate the *CNL1* and *CNL2* sequences from the *Ae. longissima* cv. TL05 reference genome sequence, with forward and reverse primers located in the 5′ untranslated region (UTR) and 3′ UTR regions, respectively. The genomic DNA sequences of *CNL1* and *CNL2* were amplified via gDNA templates from TA7548 and CS. The 30 µl PCR mixture contained 15 µl of high-fidelity enzyme 2× Phanta Flash Master Mix (Dye Plus) (Cat. No. P520, Vazyme, Nanjing, China), 2 µl of each primer (5.0 µmol/L), 4 µl of template gDNA (100 ng/µL), and 7 µl of ddH$_2$O. Amplification proceeded at 98 °C for 30 s, followed by 35 cycles of 98 °C for 10 s, 58–60 °C (depending on the primers) for 5 s, 72 °C for 5 s/kb, and a final extension at 72 °C for 1 min. A 5 µl amplification product was electrophoresed on a 1% agarose gel to estimate the product size. The remaining PCR product was purified via the MonPure Gel & PCR Purification Kit (Cat. No. MI17101, Monad Biotech Co., Ltd., Suzhou, China), cloned, and inserted into the TOPO vector (Cat. No. C602, Vazyme, Nanjing, China) and sequenced at Sangon Biotech (Shanghai) Co., Ltd. To amplify cDNAs of *CNL1* and *CNL2*, total RNA isolated from TA7548 and CS was transcribed. The complementary DNA (cDNA) was used as a template to amplify CDSs via the primers SALF-CNL1 and SALF-CNL2, and the same methods were used for gDNA sequence isolation.

### Virus-induced gene silencing

BSMV-VIGS was performed to investigate whether candidate genes mediate powdery mildew resistance in TA7548. The sequences of the candidate genes *CNL1* and *CNL2* were aligned with the reference genome sequences of CS and *Ae. longissima* cv. TL05 to select targeted fragments with less homology to CS (identity <50% and 100% nucleotide identity <21 nt) for designing VIGS primers (Supplementary Data 2). The 182 bp sequence of *CNL1* and the 210 bp sequence of *CNL2* were separately amplified from cDNA sequences and reversely cloned

and inserted into BSMV:γ vectors via homologous recombination (ClonExpress II One Step Cloning Kit, Cat. No. C112, Vazyme Biotech, Nanjing, China). Each construct of BSMV:γCNL1 and BSMV:γCNL2, together with BSMV:γ (negative control) and BSMV:γPDS (phytoene desaturase genes, positive control), was transcribed in vitro using the mMESSAGE mMACHINE T7 Kit (Cat. No. AM1344, Invitrogen, USA) according to the manufacturer's protocol. Each of the transcripts was then mixed equally with the transcripts from the BSMV:α and BSMV:β vectors before equal amounts of 2×GKP (50 mM glycine, 30 mM K$_2$HPO$_4$, pH 9.2, 1% w/v bentonite, and 1% w/v celite) were added. After the mixture was applied to the second leaves of TA7548, the seedlings were moistened to maintain 100% humidity at 23 °C in darkness for 24 h before being grown at 20–22 °C with 16 h of supplemental light. At 14 days after viral infection, the third leaf segments were taken for resistance assessment via in vitro culture on agarose medium supplemented with 2.5 mg/L 6-benzylaminopurine (6-BA) and inoculated with *Bgt* isolate E26. At 7 dpi, the infection type of each in vitro leaf segment was recorded on a 0–4 scale, with ITs of 0–2 rated as resistant and ITs of 3-4 rated as susceptible. The transcript levels of *CNL1* and *CNL2* were determined via qRT–PCR via total RNA templates extracted from infected leaves. The primer qPCR-CNL1 was used for *CNL1*, and the primer qPCR-CNL2 was used for *CNL2* and was designed based on non-VIGS targeted sequences. For silencing each candidate gene using BSMV-VIGS, at least 15 plants were challenged by BSMV vector, and the experiments were repeated three times.

### Genetic transformation of wheat

The 4296 bp CDS was amplified from the *CNL1* cDNA construct based on the *Xma*I and *Spe*I restriction sites. This CDS was subsequently cloned downstream of the *Ubi* promoter in the pWMB110-*ProUbi* construct, yielding a pWMB110-*ProUbi:CNL1* vector for genetic transformation. The resulting pWMB110-*ProUbi:CNL1* plasmid was subsequently transformed into *Agrobacterium* strain EHA105 for genetic transformation of the common wheat cultivar Fielder, generating a total of 24 transgenic T$_0$ plants. For resistance assessment of the transgenic T$_1$ lines derived from self-pollinated T$_0$ plants, the first leaf was harvested from each plant, cultured in vitro on agarose medium containing 2.5 mg/L 6-BA, inoculated with *Bgt* isolate E26, and kept in an incubator at 20 °C for 16 h light and 18 °C for 8 h in darkness. After resistance was recorded at 10 dpi, each leaf segment was used to isolate gDNA to examine *CNL1* via the primer pairs *Ubi-F* and *CNL1-R*. The second leaves from two resistant plants of each T$_1$ line were used to examine *CNL1* gene expression levels via qRT–PCR. Three biological replicates, two leaves from two resistant seedlings per replicate, were carried out for each T$_1$ line.

Transgenic wheat plants expressing *CNL1* gene driven by its native promoter *ProCNL1* were also generated. The genomic DNA of *ProCNL1* and *CNL1* was amplified from TA7548 using primers *CNL1-COM*, which incorporated *Hind*III and *Sac*I restriction sites for cloning. The digested PCR fragment was cloned into the linearized pWMB110 vector by homologous recombination (ClonExpress II One Step Cloning Kit, Cat. No. C112, Vazyme Biotech, Nanjing, China) to yield the *ProCNL1:CNL1* construct. Genetic transformation, resistance assessment and qRT-PCR followed the above-mentioned methods, except that one half a single completely expanded new leaf from each transgenic T$_0$ seedling was used for *Bgt* resistance screening and the other half for *CNL1* expression analysis.

### Evaluation of agronomic traits

The agronomic traits of CS-*Ae. longissima* recombinant T3012II-3 containing *Pm6Sl* in the smallest 6S$^l$ segment and CS were evaluated to determine whether any linkage drag was associated with the *Pm6Sl* or 6S$^l$ segment. T3012II-3 and CS were planted in experimental fields in Beijing, China. A total of 21 plants for each plant material were planted in a 2 × 0.3 m row. Field management practices, including irrigation, fertilization, and herbicide and pesticide applications, strictly followed

local practices. Ten plants were randomly selected from each row to measure plant height (from the ground to the tip of the main spike, excluding awns), tiller number per plant and spike number per plant. Three of the early-tillering spikes from each plant were then measured to determine spike length (the length from the base of the rachis to the uppermost spikelet top excluding awns), number of spikelets per spike, number of grains per spike, grain length, grain width, and one thousand-grain weight. The significant differences in agronomic traits between T3012II-3 and CS were determined for each trait via two-tailed Student's $t$ test.

## H$_2$O$_2$ accumulation and wheat cell death assays

To investigate the differential accumulation of hydrogen peroxide (H$_2$O$_2$) in T3012II-3 cells mediated by *Pm6Sl* and CS lacking *Pm6Sl* following pathogen infection, we used DAB staining to observe intracellular reactive oxygen species (intraROS) in infected cells. To do so, leaves infected with *Bgt* isolate E26 were treated with a 0.1% (w/v) solution of DAB (pH 3.8) at 25 °C in darkness for 12 h. The leaves were then decolorized in a solution of ethanol and acetic acid (3:1, v/v) at room temperature until complete decolorization was achieved. Finally, the decoloured leaves were stained with a 0.6% (w/v) solution of Coomassie brilliant blue for 10 s before being rinsed with water to visualize the *Bgt* fungal structure for observation.

To quantify cell death levels in infected cells, leaves inoculated with *Bgt* isolate E26 were obtained from CS and TA7548 at 48 hpi and then immersed in a 0.4% trypan blue solution that was brought to the boiling point for exactly one minute. After immersion, the leaves were bleached for 24 h with chloral hydrate solution containing 2.5 g/ml and then stained with a 0.6% (w/v) Coomassie blue solution for 10 s. The treated leaves were observed under a Nikon ECLIPSE Ni-U microscope. All the experiments were carried out with three biological replicates, each with a single leaf. A minimum of 100 infected cells were assessed per leaf to calculate the proportions of cells exhibiting intraROS and cell death responses[52].

## Prediction of structural domains and 3D structures

The websites SMART, NCBI and InterPro (http://www.ebi.ac.uk/interpro/search/sequence/) were used to predict Pm6Sl structural domains. The MPI Bioinformatics Toolkit (https://toolkit.tuebingen.mpg.de/tools/deepcoil2) was used to predict the CC domain of Pm6Sl, and the software "ColabFold v1.5.2: AlphaFold2 in using MMseqs2" was used to predict its three-dimensional (3D) construction. Five structural models were predicted in total, and the first-ranked model was selected for structural analysis. The protein sequences and domain compositions of Yr5, Yr7, Xa1, and Rph15 are referenced from published articles. We individually extracted the sequence before the NB-ARC domain of each protein and used Alphafold2 to predict their 3D structures. PyMOL 2.6.0a0 was used to open the 3D structural model and polish and mark the model.

## *Agrobacterium*-mediated transient expression in *N. benthamiana*

To investigate whether individual or combined domains of Pm6Sl can induce cell death, expression vectors containing various truncated forms of Pm6Sl were generated for transient expression in *N. benthamiana*. The different truncated *Pm6Sl* sequences were amplified using primers listed in Supplementary Data 2 and cloned between the double *35S* promoter and GFP in the pCambia 1305.1-GFP vector based on *Spe*I and *BamH*I restriction sites. This resulted in the construction of pCambia 1305.1-truncated Pm6Sl-GFP vectors, where GFP was fused to the C-terminus of truncated Pm6Sl proteins. The truncated sequences of the α1–4 (CC), Znf-BED, and α1-3-BED-α4 (CC-BED) modules of Yr5, Yr7, Xa1 and Rph15 were synthesized by Sangon Biotech (Shanghai) Co., Ltd., on the basis of published gene sequences. The pCambia 1305.1-truncated Yr7/Yr5/Xa1/Rph15-GFP vectors were constructed

using the same method used for Pm6Sl. These vectors were subsequently transformed into *Agrobacterium* strain GV3101, using the pCambia 1305.1-GFP vector as a negative control and pCambia 1305.1-Pm21CC-GFP causing *N. benthamiana* cell death as a positive control[53]. After supplementation with 10 mmol/L MES and 20 µmol/L acetosyringone, the activated *Agrobacterium* strain was cultured until the OD$_{600}$ value reached ~1.0. *Agrobacterium* cells were subsequently collected via centrifugation and resuspended in an injection buffer containing 10 mmol/L MgCl$_2$, 10 mmol/L MES, and 100 µmol/L acetosyringone to adjust the OD$_{600}$ value in the range of 1.0–1.5. Following a 3-h incubation at room temperature, each *Agrobacterium* strain containing a distinct plasmid carrying one truncated form was injected into a circular area of 1 cm diameter on one leaf of an *N. benthamiana* plant at the 4–6-leaf stage. After 3 d of culture at 22 °C under a light source, the phenotype of the injected area was recorded, and TPN staining was conducted to confirm the cell death phenotype[54]. Proteins transiently expressed in *N. benthamiana* were detected via western blotting using an anti-GFP antibody (1:5000, QiYan, Beijing, China, No.QYA03914A).

## Phylogenetic analysis

Homologues of Pm6Sl were identified via the Homologous function in the TGT System (http://wheat.cau.edu.cn/TGT/index.html)[55] and NCBI BLAST. A phylogenetic tree of Pm6Sl was constructed along with 180 other proteins from the Gramineae family used to analyze the CNL network[56], as well as six other BED-NLRs with resistance to diverse pathogens, including Rph15, Yr7, Yr5, YrSP, Xa1 and Xo1. For phylogenetic analysis of the Znf-BED domains, Pm6Sl and six other crop BED-NLRs, along with 154 non-NLR and NLR proteins containing Znf-BED domains in the Gramineae family reported by Marchal *et al*[57]. were used. For phylogenetic analysis of the Pm6Sl and CC-BED modules, six reported crop BED-NLRs, along with 46 other BED-NLRs in the Gramineae family, were used. Multiple sequence alignment was performed via MAFFT v7.475 L-INS-I (https://mafft.cbrc.jp/alignment/software/) with default parameters. A phylogenetic tree was constructed via the approximate maximum likelihood method via FastTree v2.1.11 (http://www.microbesonline.org/fasttree/). The tree was visualized via iTOL (https://itol.embl.de/)[58].

## Development and validation of a diagnostic gene marker

A gene marker, *pm6sl-1*, for *Pm6Sl* was designed based on genomic DNA sequences of *Pm6Sl* in comparison with wheat genomic reference sequences collected from WheatOmics 1.0 (http://wheatomics.sdau.edu.cn/)[59]. PCR amplification was conducted via the primer pair 5'-GCCGAGCTAACAACCTTCCTC-3'/5'-GCATGGGAATTAATGTTGTG-3' for *pm6sl-1*, resulting in amplicons of 904 bp. To determine the diagnostic value of the marker, *Pm6Sl* stocks, including TA7548, TA7548S and CS-*Ae. longissima* recombinant T3012II-3, along with a selection of 112 common wheat varieties, were used (Supplementary Data 8). The amplification products were separated on 1.0% agarose gels, stained with 4SGelred nucleic acid dye (Cat. No. A616697, BBI Life Science Corporation, Shanghai, China), and detected using a GBOXSYDR4/1494 Syngene Automated Gel Imager (Syngene, Frederick, MD21704, USA) under UV light.

## Reporting summary

Further information on research design is available in the Nature Portfolio Reporting Summary linked to this article.

# Data availability

All data generated or analyzed in this study are included in this article and Supplementary Information files as well as the public databases. The detailed sequence data of *Pm6Sl* are accessible on the NCBI website under accession OR736058. Source data are provided with this paper.

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

## Acknowledgements

We are grateful to Prof. Guihua Bai from Kansas State University, Manhattan, KS, USA for amending this manuscript and valuable suggestion. This research was financially supported by the National Natural Science Foundation of China (32372089 to W.X.L., 32401805 to Q.W.L. and 32272070 to H.H.L.), the Major Science and Technology Projects of Henan Province (221100110700 to G.H.Y.), the Natural Science Foundation of Henan (242300420489 to Q.W.L.), the Graduate Education Reform Project of Henan Province (2023SJGLX052Y to Y.Z.) and the China Postdoctoral Science Foundation (2024M750806 to Q.W.L.).

## Author contributions

Q.W.L., Y.S.Z., W.X.L. and Y.Z. designed the study. C.M., X.B.T., Z.J.D., H.H.L., X.X.C., G.H.Y., S.Y.M., L.W.Z., H.F.Y. and S.W.W. performed the research. Y.Z., Z.B.Z., C.L. and S.S. analyzed the data. W.X.L., X.B.T., C.M. and Y.S.Z. wrote the manuscript and S.S., Y.Z., A.Z.C. and B.L. contributed to revising the draft. All authors have read and approved the final manuscript. All authors have read and approved the final manuscript.

## Competing interests

The authors declare no competing interests.
