## [Peer Review File · Nature Communications]

An *Aegilops longissima* NLR protein with integrated CC-BED module mediates resistance to wheat powdery mildewReviewers' Comments:

Reviewer #1:

Remarks to the Author:

In their manuscript, Ma et al. utilize a classical map-based cloning approach to identify and clone Pm6Sl, a CC-NLR gene responsible for conferring resistance against the wheat powdery mildew disease *Blumeria graminis* f. sp. *tritici* (Bgt). To further narrow down their candidate genes, the authors conducted extensive EMS mutagenesis, identifying over 30 individual mutants affected in the Pm6Sl gene all showing loss of the resistance phenotype. Subsequently, they employed both transient silencing of Pm6Sl using BSMV-VIGS and stable transgenic transformation of Pm6Sl into hexaploid wheat to validate the role of Pm6Sl in conferring resistance against Bgt.

Notably, the authors show that the Pm6Sl resistance is mediated by classical hypersensitive-response-. However Pm6Sl stands-out among previously cloned CC-NLR active against powdery mildew in containing a Zinc finger-like integrated domain. The authors also make intriguing observations that the CC-BED domain of Pm6Sl, unlike that of other CC-BED modules of other NLR, triggers a hypersensitive response in *Nicotiana benthamiana*. Given the unconventional structure of Pm6Sl, this research holds broader significance in providing the basis to unravel the functional significance of the CC-BED modules. .

To validate the role of the CC-BED module in Pm6Sl-mediated cell death and to substantiate the findings outlined in the manuscript, I propose the of the following two experiments:

1. Expression of the full-length Pm6Sl in *Nicotiana benthamiana* to evaluate its capacity to induce a cell death response.
2. Assessment of the functional consequences of the mutations identified in the CC-BED modules through EMS mutagenesis, particularly regarding their impact on cell death induction in the *Nicotiana* assay.

In addition, the following points should be addressed prior to publication:

Does the reference line, TL05 contain the active allele of Pm6Sl?

Please provide loading controls of your western blots. (Commassie stained Rubisco band or equivalent).

In the methods section it is written, that each leaf segment in the VIGS assay was scored individually: please provide this data as a figure to show the variability of your VIGS assay.

When phenotype transgenic lines expressing a dominant R gene in the T1 generation, you expect your transgene to still segregate in the individual T1 families. Was this the case for the analysed T1 families?

Furthermore, the discussion section could benefit from a more comprehensive analysis of relevant literature regarding the mode of action of the BED module in other pathosystems. For instance, exploring experimental evidence concerning Xa1

(<https://nph.onlinelibrary.wiley.com/doi/full/10.1111/nph.18439>) Additionally, it may be noteworthy to mention recent findings demonstrating the targeting of a transcription factor with a zinc finger fold by a Bgt effector (reference: [https://www.cell.com/plant-communications/pdfExtended/S2590-3462\(23\)00327-9](https://www.cell.com/plant-communications/pdfExtended/S2590-3462(23)00327-9)).

Minor points:

Typo in Figure 2c: Negaitve

Reviewer #2:

Remarks to the Author:

Title: An *Aegilops longissima* NLR protein with integrated CC-BED module 1 mediates resistance to wheat powdery mildew

Authors: Ma et al.

Ms #: Nature Plants: NCOMMS-24-14168

Powdery mildew (PM) of wheat, caused by the fungus *Blumeria graminis* f. sp. *tritici*, is a widespread and damaging disease of wheat in many production areas across the world. Many different loci and alleles for PM resistance (Pm genes) have been described in the primary, secondary, and tertiary gene pools of wheat. In this investigation, the authors identified a novel PM resistance gene, provisionally designated Pm6Sl, on chromosome 6Sl#3 from *Aegilops longissima*, a member of the Sitopsis section in the secondary gene pool of wheat. They subsequently transferred the gene to bread wheat based on homoeologous recombination between 6Sl#3 and wheat counterparts induced by ph1b gene. The authors then went on to isolate Pm6Sl by combining map-based cloning with ph1b-induced homologous recombination, and candidate gene sequence comparison of loss-of-function mutants and the Pm6Sl donor.

Although 18 different PM resistance genes have already been cloned from wheat and its wild relatives, Pm6Sl encodes a nucleotide-binding leucine-rich repeat (NLR) protein possessing a CC-BED module, which is formed by an additional Zinc finger BED (Znf-BED) domain integrated into the coiled-coil (CC) domain. This is a novel protein among cloned wheat PM resistance genes. The mechanism regarding the involvement of the Znf-BED domain in pathogen resistance of plant NLRs warrants further investigation.

The function of Pm6Sl was validated using three different approaches: mutagenesis, gene silencing, and transgenic assays. In addition, the authors developed germplasm lines that carried Pm6Sl, but had very little of the residual donor chromatin remaining from *Aeg. longissima*. Finally, the authors developed a diagnostic marker to facilitate the transfer of Pm6Sl in breeding programs. These latter achievements will facilitate the use of the PM resistance gene in breeding programs.

Overall, the research presented in this study was comprehensive, using rigorous methodology for the cloning and validation of Pm6Sl. Other ancillary studies contributed to our understanding of this novel PM resistance gene. The investigators characterized the structural characteristics of the Pm6Sl protein and compared it to other reported ones with a Znf-BED domain. They also studied how Pm6Sl may confer resistance by assaying cells for cell death following inoculation with *B. graminis* f. sp. *tritici* isolate E26. Through a search of Pm6Sl homologs in the NCBI non-redundant protein sequence database, no close hit was found indicating the protein was only found in *Aeg. longissima*. The authors also evaluated recombinant wheat stocks with Pm6Sl and compared their

performance to the wild type of Chinese Spring. No significant differences were found for yield or other yield parameters in their field studies. They also demonstrated that Pm6Sl conferred all-stage resistance to PM.

This research is a major contribution toward our understanding of novel R gene motifs in disease resistance. It also contributed advanced germplasm with PM resistance that can be readily used by breeders. Moreover, the manuscript itself was well written and easy to follow. There were a few passages here and there that were not in proper idiomatic English. I have highlighted a few of them below.

For the passage in lines 57-59, there is this small wording change:

These include 12 (66.7%) Pm genes, including as Pm1a, Pm2, Pm3b/Pm8/Pm17, Pm5e, Pm12/Pm21, Pm41, Pm55, Pm60 and Pm69, encoding nucleotide-binding leucine-rich repeat (NLR) proteins, one gene each encoding an ATP-binding cassette (ABC) transporter (Pm38) and a hexose transporter (Pm46).

For the passage in lines 66-67, there is this small wording change:

However, more effort is needed to understand the diverse array of complex mechanisms that activate plant resistance.

For the passage in line 98, there is this small wording change:

“...mildew resistance gene, temporarily designated as Pm6Sl...” You could also say “provisionally”

Question regarding the passages in lines 98-100:

What was the origin (site of collection and genebank holding the seed) of the *Ae. longissima* accession donor TA7548 used in the study? How widely effective is Pm6S against an array of isolates of *Blumeria graminis* f. sp. *tritici*? Was this accession widely resistant against the pathogen to warrant the effort in cloning and also transfer into wheat?

For the passage in lines 109-112, there is this small wording change:

This study documents a novel resistant gene for elucidating the molecular mechanisms underlying gene-for-gene interaction between wheat and its pathogens. Moreover, new wheat lines and a diagnostic marker for rapid deployment of Pm6Sl were developed to facilitate powdery mildew resistance breeding in wheat.

For the heading in line 246, there is this small wording change:

Pm6Sl is only present in *Aegilops longissima*

Question regarding the passages in lines 467-468:

It is indicated that detached leaf segments were used in the powdery mildew assays with VIGS. Why not test whole (intact) plants for their disease reaction and subsequent gene expression levels? This requires only to plant the seed and grow the plants instead of having to cut leaves and artificially place them on the media for the detached leaf assays.

Question regarding the passages in lines 496-499:

You indicate that “Three of the largest spikes from each plant were then measured to determine spike length.” Why weren’t the spikes randomly selected for this assay? Is there an inherent bias in the sampling method you employed?

Reviewer #3:

Remarks to the Author:

The manuscript reports on the isolation of the encoding gene, Pm6Sl present in a small introgression segment of *Aegilops longissima* originally introduced into wheat as a whole chromosome addition and subsequently substitution. The resistance phenotype of Pm6Sl was assayed with a single *Blumeria graminis* fsp *tritici* (Bgt) isolate E26 and formed the basis of assessment of mutants, VIGS and transgene analysis. The large number of independent mutants within a single CNL with an integrated BED domain provided strong evidence for the isolation of the causal gene and subsequently verified with VIGS and transgene (albeit with an overexpression promoter) complementation. Comparisons are made with other CNL’s with an integrated BED domain which revealed similar structures but differing in a transient cell death assay in a heterologous system for which the authors make claims to an “atypical NLR” encoded by Pm6Sl. The authors further demonstrate the potential value of the small introgressed segment from the absence of any deleterious (linkage drag) effects on a range of agronomic traits.

Major comments:

1) The evidence for the CNL with an integrated BED domain as the causal gene for Pm6Sl is very solid and well documented in the manuscript. The main criticism of the paper is the repeated claims to the “unique”, “novel” and “atypical NLR” based on the CC-BED integrated domain. First of all the structure of the integrated BED domain in CNL’s have been previously reported for various pathosystems in wheat, barley and rice and as such there is nothing atypical or novel to the claims being made in the current paper. Essentially, they are dealing an NLR with an integrated domain, something that has been well established in the literature. Secondly, reliance on a transient cell death assay in *N benthamiana*, a heterologous system, to distinguish the CC-BED domain of Pm6Sl from other cereal CC-BED domain effects are not the grounds to claim atypical NLR characteristics. A more convincing approach requires the tests to be conducted in the organisms harboring the targeted genes where their interacting partners are expected to be present. Consequently, all sections of the manuscript referring to “unique”, “novel” and “atypical NLR” features of Pm6Sl needs to be revised and such descriptors removed.

2) It is puzzling that after a detailed EMS mutagenesis study that clearly identifies the candidate gene, the authors choose to use an over-expression promoter (ubi) and the cDNA of the CNL candidate as their gene construct for the transgenic complementation. Over-expression constructs can mask pathogen specificity tests. Why was the native promoter and the full genomic sequence of the encoding region not used in the transgenic studies? The authors can at least provide this data and can be compared with the overexpressing Pm6Sl-cDNA construct.

3) What is missing from the study is to what extent Pm6Sl confers resistance to multiple Bgt isolates as the authors make claims to its breeding value. I can only see data presented for a single Bgt isolate. This information is critical in defining the usefulness of Pm6Sl. Previous studies about the spectrum of isolates on the chromosome harbouring Pm6Sl was based on the full chromosome addition line. With the availability of the smallest introgression segment with Pm6Sl, the authors should provide data using this genetic stock rather than relying on previous addition/full chromosome substitution line. The transgenic line using the over-expressing construct will be inappropriate. (The authors will be well aware of the previous Pm21 over-expression with a candidate kinase that was later corrected when it became evident that Pm21 encoded a CNL).

4) Again, it is noted that the whole chromosome addition line (TA7548) was what was used in the cell death and intracellular reactive oxygen species studies shown in Figure 3 and described in the Results. The authors should provide TPN-Coomassie blue and DAB-Coomassie blue data using the small introgressed segment harbouring Pm6Sl for any claims attributed to the Pm6Sl gene. Similarly with Figure 2, the powdery mildew resistance phenotype should show what can be unequivocally attributed to Pm6Sl by using recombinant T3012II-3 containing Pm6Sl in the smallest 6Sl segment, rather than TA7548. After all the emphasis of the manuscript is about the isolation of the Pm6Sl gene not the original chromosome addition (TA7548) into wheat which has previously been published.

Other comments:

The section of the manuscript entitled “Pm6Sl is only present in *Aegilops longissima*” is rather premature. Currently the database of wheat relatives from the Sitopsis section is limited and hence any claims to Pm6Sl being confined to *Ae longissimi* should await more genome sequences for diverse accessions of the Sitopsis (S genome) species. Consequently, the heading in line 246 and statements in lines 251-252 needs to be revised.

The manuscript will benefit from an English revision. Some glaring areas include:

Line 57 insert ...Pm69 ‘which’ encode

Line 66-67 rephrase the sentences

Line 98 replace temperately with temporarily

Line 111 replace presents with represents

Line 129-130 Rephrase the sentence

Line 376 insert were between leaves and fully

Line 420-421, remove were from seedling and segregated

REVIEWER COMMENTS

Reviewer #1 (Remarks to the Author):

In their manuscript, Ma et al. utilize a classical map-based cloning approach to identify and clone Pm6SI, a CC-NLR gene responsible for conferring resistance against the wheat powdery mildew disease *Blumeria graminis* f. sp. *tritici* (Bgt). To further narrow down their candidate genes, the authors conducted extensive EMS mutagenesis, identifying over 30 individual mutants affected in the Pm6SI gene all showing loss of the resistance phenotype. Subsequently, they employed both transient silencing of Pm6SI using BSMV-VIGS and stable transgenic transformation of Pm6SI into hexaploid wheat to validate the role of Pm6SI in conferring resistance against Bgt.

Notably, the the authors show that the Pm6SI resistance is mediated by classical hypersensitive-response-. However Pm6SI stands-out among previously cloned CC-NLR active against powdery mildew in containing a Zinc finger-like integrated domain. The authors also make intriguing observations that the CC-BED domain of Pm6SI, unlike that of other CC-BED modules of other NLR, triggers a hypersensitive response in *Nicotiana benthamiana*. Given the unconventional structure of Pm6SI, this research holds broader significance in providing the basis to unravel the functional significance of the CC-BED modules.

We thank you for your comments and positive assessment of our manuscript.

To validate the role of the CC-BED module in Pm6SI-mediated cell death and to substantiate the findings outlined in the manuscript, I propose the of the following two experiments:

1. Expression of the full-length Pm6SI in *Nicotiana benthamiana* to evaluate its capacity to induce a cell death response.

>Response: As per your suggestion, we incorporated cell death induction analysis of the full-length Pm6SI in *Nicotiana benthamiana* within Supplementary Fig. 6 in reversed version. Specifically, while expressing the full-length Pm6SI in *Nicotiana benthamiana*, we observed that, similar to the reported results with the BED-NLR protein Yr7 conferring resistance to wheat yellow leaf rust (Marchal *et al.*, 2020), it failed to induce cell death in *Nicotiana*.

Marchal, C., Haberer, G., Spannagl, M. & Uauy, C. Comparative genomics and functional studies of wheat BED-NLR loci. *Genes (Basel)* **11**, 1460 (2020).

2. Assessment of the functional consequences of the mutations identified in the CC-BED modules through EMS mutagenesis, particularly regarding their impact on cell death induction in the *Nicotiana* assay.

>Response: We assayed the functional consequences of two mutations in Pm6SI CC-BED modules (Mutant 28 and 29), each in the CC subdomain and the BED subdomain. Mutant CC-BED modules and control CC-BED module of Pm6SI all induced cell death in *Nicotiana* (Fig. 5b), indicating no impact of Pm6SI CC-BED module mutations in *Nicotiana*

cell death induction, although both mutations changed Pm6SI from powdery mildew resistance to susceptibility. The evaluation confirmed our previous finding that of the Pm6SI, Yr5/Yr7/YrSP, Xa1 and Rph15 CC-BED modules, only Pm6SI could induce *Nicotiana* cell death. Furthermore, it was suggested that these two mutations resulted in the loss of Pm6SI resistance most likely by influencing other processes, such as pathogen recognition, rather than directly affecting cell death induction.

In addition, the following points should be addressed prior to publication:

Does the reference line, TL05 contain the active allele of Pm6SI?

>Response: We assayed the powdery mildew resistance of TL05 and found that TL05 exhibits resistance against *Bgt* isolate E26. Furthermore, a comparative analysis of amino acid sequences indicated that the TL05 Pm6SI allele shares a high degree of similarity with Pm6SI (Identity = 99.4%). This indicates that TL05 probably contains the active allele of Pm6SI. Nonetheless, several powdery mildew resistance genes, including *Pm13* at chromosome 3S¹ and *Pm66* at 4S¹, originated from *Aegilops longissima*, making it uncertain whether the *Pm6SI* allele in TL05 contributes to resistance or not.

Please provide loading controls of your western blots. (Commassie stained Rubisco band or equivalent).

>Response: We have provided loading controls in western blots as shown in Fig. 5.

In the methods section it is written, that each leaf segment in the VIGS assay was scored individually: please provide this data as a figure to show the variability of your VIGS assay.

>Response: In Fig. 2b, we present the results of the VIGS assay on three leaf segments from three different plants, together with their corresponding expression levels. As evident from the figure, the BSMV:γCNL1-1 leaf segment displays the highest susceptibility to the disease, and correspondingly, the expression level of the *CNL1* gene in this segment is the lowest.

When phenotype transgenic lines expressing a dominant R gene in the T1 generation, you expect your transgene to still segregate in the individual T1 families. Was this the case for the analysed T1 families?

>Response: Yes. Of the 24 transgenic T₁ families, 17 were segregated for both transgenic *Pm6SI* and resistance phenotype, and all *Pm6SI*-positive plants displayed resistance to powdery mildew. Three T₁ families positive for *Pm6SI* (L1, L3, and L12) displayed a uniform resistance phenotype with no segregation. The remaining four T₁ families (L13, L14, L15, and L24) were *Pm6SI*-negative and susceptible to powdery mildew. We have rephrased this statement in our results section to emphasize that individual plants carrying *CNL1* in T₁ transgenic families exhibit resistance to *Bgt* isolate E26 (see Lines 188-196).

Furthermore, the discussion section could benefit from a more comprehensive analysis of relevant literature regarding the mode of action of the BED module in other pathosystems. For instance, exploring experimental evidence concerning Xa1 (<https://nph.onlinelibrary.wiley.com/doi/full/10.1111/nph.18439>) Additionally, it may be

noteworthy to mention recent findings demonstrating the targeting of a transcription factor with a zinc finger fold by a Bgt effector (reference: [https://www.cell.com/plant-communications/pdfExtended/S2590-3462\(23\)00327-9](https://www.cell.com/plant-communications/pdfExtended/S2590-3462(23)00327-9)).

>Response: Thank you for your thoughtful comments and suggestions, which have greatly contributed to enhancing the quality and depth of our manuscript. We have thoroughly revised the Discussion section to include a more comprehensive analysis of the relevant literature on the mode of action of BED across diverse pathosystems (see Lines 365-384).

Minor points:

Typo in Figure 2c: Negaitve

>Response: Thanks! We have corrected to “Negative”.

Reviewer #2 (Remarks to the Author):

Title: An *Aegilops longissima* NLR protein with integrated CC-BED module 1 mediates resistance to wheat powdery mildew

Authors: Ma et al.

Ms #: Nature Plants: NCOMMS-24-14168

Powdery mildew (PM) of wheat, caused by the fungus *Blumeria graminis* f. sp. *tritici*, is a widespread and damaging disease of wheat in many production areas across the world. Many different loci and alleles for PM resistance (Pm genes) have been described in the primary, secondary, and tertiary gene pools of wheat. In this investigation, the authors identified a novel PM resistance gene, provisionally designated Pm6SI, on chromosome 6SI#3 from *Aegilops longissima*, a member of the *Sitopsis* section in the secondary gene pool of wheat. They subsequently transferred the gene to bread wheat based on homoeologous recombination between 6SI#3 and wheat counterparts induced by ph1b gene. The authors then went on to isolate Pm6SI by combining map-based cloning with ph1b-induced homologous recombination, and candidate gene sequence comparison of loss-of-function mutants and the Pm6SI donor.

Although 18 different PM resistance genes have already been cloned from wheat and its wild relatives, Pm6SI encodes a nucleotide-binding leucine-rich repeat (NLR) protein possessing a CC-BED module, which is formed by an additional Zinc finger BED (Znf-BED) domain integrated into the coiled-coil (CC) domain. This is a novel protein among cloned wheat PM resistance genes. The mechanism regarding the involvement of the Znf-BED domain in pathogen resistance of plant NLRs warrants further investigation.

The function of Pm6SI was validated using three different approaches: mutagenesis, gene silencing, and transgenic assays. In addition, the authors developed germplasm lines that

carried Pm6SI, but had very little of the residual donor chromatin remaining from *Aeg. longissima*. Finally, the authors developed a diagnostic marker to facilitate the transfer of Pm6SI in breeding programs. These latter achievements will facilitate the use of the PM resistance gene in breeding programs.

Overall, the research presented in this study was comprehensive, using rigorous methodology for the cloning and validation of Pm6SI. Other ancillary studies contributed to our understanding of this novel PM resistance gene. The investigators characterized the structural characteristics of the Pm6SI protein and compared it to other reported ones with a Znf-BED domain. They also studied how Pm6SI may confer resistance by assaying cells for cell death following inoculation with *B. graminis* f. sp. *tritici* isolate E26. Through a search of Pm6SI homologs in the NCBI non-redundant protein sequence database, no close hit was found indicating the protein was only found in *Aeg. longissima*. The authors also evaluated recombinant wheat stocks with Pm6SI and compared their performance to the wild type of Chinese Spring. No significant differences were found for yield or other yield parameters in their field studies. They also demonstrated that Pm6SI conferred all-stage resistance to PM.

This research is a major contribution toward our understanding of novel R gene motifs in disease resistance. It also contributed advanced germplasm with PM resistance that can be readily used by breeders. Moreover, the manuscript itself was well written and easy to follow. There were a few passages here and there that were not in proper idiomatic English. I have highlighted a few of them below.

We thank you for your comments and positive assessment of the manuscript.

For the passage in lines 57-59, there is this small wording change:

These include 12 (66.7%) Pm genes, including as Pm1a, Pm2, Pm3b/Pm8/Pm17, Pm5e, Pm12/Pm21, Pm41, Pm55, Pm60 and Pm69, encoding nucleotide-binding leucine-rich repeat (NLR) proteins, one gene each encoding an ATP-binding cassette (ABC) transporter (Pm38) and a hexose transporter (Pm46).

>Response: Thank you! We have rephrased the passage as you suggested (see Lines 55-58). We also enhanced the manuscript's language quality with editing from Springer Nature Author Services.

For the passage in lines 66-67, there is this small wording change:

However, more effort is needed to understand the diverse array of complex mechanisms that activate plant resistance.

>Response: Thank you! We have revised this sentence as you suggested (see Lines 66-67).

For the passage in line 98, there is this small wording change:

"...mildew resistance gene, temporarily designated as Pm6SI..." You could also say "provisionally"

>Response: We have corrected it (see Line 99).

Question regarding the passages in lines 98-100:

What was the origin (site of collection and genebank holding the seed) of the *Ae. longissima* accession donor TA7548 used in the study? How widely effective is *Pm6S1* against an array of isolates of *Blumeria graminis* f. sp. *tritici*? Was this accession widely resistant against the pathogen to warrant the effort in cloning and also transfer into wheat?

>Response: TA7548 was provided by Wheat Genetics Resource Center (WGRC), Kansas State University, USA (https://wgrc.k-state.edu/database_search/?table=genetic_stocks&stock_type=Alien%20Addition&page_no=6). TA7548 is a Chinese Spring (CS)-*Ae. longissima* disomic 6S¹ addition line (2n=44), where a pair of *Ae. longissima* chromosome 6S¹ was added to the CS background. The 6S¹ donor, *Ae. longissima* accession TA1910 (2n=14), is provided by Texas A&M University and also maintained at WGRC. Its recorded acquisition date is July 30, 1911, and it was collected in Israel, although the precise location is unknown (<https://www.genesys-pgr.org/a/694e93bd-b8af-4d27-a66d-a04f51de55df>).

Previously, we evaluated the resistance of 6S¹ addition line TA7548 using 30 *Bgt* isolates collected from various major wheat growing regions in China. As a result, the 6S¹ harboring *Pm6S1* conferred resistance (ITs = 0, 1, 2) to 28 of these isolates (Tian *et al.*, 2022). In the revised manuscript, following your suggestion, we further performed a resistance spectrum analysis of *Pm6S1* using recombinant T3012II-3, which carries *Pm6S1* in the smallest 6S¹ segment. The results show that the resistance of T3012II-3 is consistent with the 6S¹ addition line TA7548, being resistant to 35 of the 36 single-spore derived *Bgt* isolates tested from diverse wheat cultivation regions in China. Thus, *Pm6S1* provides a broad spectrum of resistance to wheat powdery mildew.

In addition, *Pm6S1* confers all-stage resistance, making it worthwhile for cloning and transferring it to wheat for breeding purposes. However, *Pm6S1* is not immune to all tested *Bgt* isolates. It would also be beneficial for us to identify its effector factors and further dissect its disease resistance mechanism.

Tian, X. *et al.* Development and characterization of *Triticum aestivum*-*Aegilops longissima* 6S¹ recombinants harboring a novel powdery mildew resistance gene *Pm6S1*. *Front. Plant Sci.* **13**, 918508 (2022).

For the passage in lines 109-112, there is this small wording change:

This study documents a novel resistant gene for elucidating the molecular mechanisms underlying gene-for-gene interaction between wheat and its pathogens. Moreover, new wheat lines and a diagnostic marker for rapid deployment of *Pm6S1* were developed to facilitate powdery mildew resistance breeding in wheat.

>Response: We have revised the passage as you suggested (see Lines 110-114).

For the heading in line 246, there is this small wording change:

Pm6S1 is only present in *Aegilops longissima*

>Response: Following your suggestion, we have revised the heading to " Pm6SI is likely present exclusively in *Aegilops longissima*" (see Line 276).

Question regarding the passages in lines 467-468:

It is indicated that detached leaf segments were used in the powdery mildew assays with VIGS. Why not test whole (intact) plants for their disease reaction and subsequent gene expression levels? This requires only to plant the seed and grow the plants instead of having to cut leaves and artificially place them on the media for the detached leaf assays.

>Response: Powdery mildew resistance assessment of a wheat plant can be performed using either live seedlings or detached leaf segments cultured *in vitro* on medium containing 6-BA, both yielding satisfactory results. When conducting resistance phenotype experiments on hot summer days, such as VIGS and transgenic plant phenotyping, we chose the detached leaf assay due to its minimal space requirement, facilitating maximally consistent experimental environment and inoculated pathogen spore amounts. Leaf segments are detached from the same leaf of each plant (typically the third leaf), and are crisp on media for uniform *Bgt* spore inoculation.

Question regarding the passages in lines 496-499:

You indicate that "Three of the largest spikes from each plant were then measured to determine spike length." Why weren't the spikes randomly selected for this assay? Is there an inherent bias in the sampling method you employed?

>Response: Sorry, 'the largest spikes' was inaccurately described because we actually selected three spikes from three early tillers at the early tillering stage. That 'the largest spikes' has been revised to ' early-tillering spikes' in the revised manuscript (see Line 558). When investigating spike traits such as spike length, spikelet number, and grain number per spike, the main spike of a plant is usually used. However, since Chinese Spring (CS) and CS-*Ae. longissima* addition lines or recombinant lines grew many tillers (>20 per plant) in our experimental fields, it is very difficult to distinguish the main spike from other early tiller spikes. Therefore, we selected three of the early tiller spikes from each plant to measure and calculate the average spike length to mitigate sampling variance. Despite selecting three early tiller spikes per plant, scientific randomness was maintained by randomly choosing the plants for spike length measurement, eliminating any inherent bias in our sampling procedure.

Reviewer #3 (Remarks to the Author):

The manuscript reports on the isolation of the encoding gene, Pm6SI present in a small introgression segment of *Aegilops longissima* originally introduced into wheat as a whole chromosome addition and subsequently substitution. The resistance phenotype of Pm6SI was assayed with a single *Blumeria graminis* fsp tritici (*Bgt*) isolate E26 and formed the basis of assessment of mutants, VIGS and transgene analysis. The large number of independent mutants within a single CNL with an integrated BED domain provided strong

evidence for the isolation of the causal gene and subsequently verified with VIGS and transgene (albeit with an overexpression promoter) complementation. Comparisons are made with other CNL's with an integrated BED domain which revealed similar structures but differing in a transient cell death assay in a heterologous system for which the authors make claims to an "atypical NLR" encoded by Pm6SI. The authors further demonstrate the potential value of the small introgressed segment from the absence of any deleterious (linkage drag) effects on a range of agronomic traits.

We thank you for your comments and positive assessment of the manuscript.

Major comments:

1) The evidence for the CNL with an integrated BED domain as the causal gene for Pm6SI is very solid and well documented in the manuscript. The main criticism of the paper is the repeated claims to the "unique", "novel" and "atypical NLR" based on the CC-BED integrated domain. First of all the structure of the integrated BED domain in CNL's have been previously reported for various pathosystems in wheat, barley and rice and as such there is nothing atypical or novel to the claims being made in the current paper. Essentially, they are dealing an NLR with an integrated domain, something that has been well established in the literature. Secondly, reliance on a transient cell death assay in *N benthamiana*, a heterologous system, to distinguish the CC-BED domain of Pm6SI from other cereal CC-BED domain effects are not the grounds to claim atypical NLR characteristics. A more convincing approach requires the tests to be conducted in the organisms harboring the targeted genes where their interacting partners are expected to be present. Consequently, all sections of the manuscript referring to "unique", "novel" and "atypical NLR" features of Pm6SI needs to be revised and such descriptors removed.

>Response: Thank you for your valuable suggestions. We apologize for the inadequate claims we made, such as "unique", "novel", and "atypical NLR". In some literature, nucleotide-binding leucine-rich repeat receptors (NLRs) containing integrated domains (IDs), such as WRKY, kinase, and BED finger domains, are called atypical (non-canonical) NLRs. We call Pm6SI by atypical NLR only to mean that it is a NLR with an ID domain. In any case, we agree with you that the claims of "unique", "novel" and "atypical NLR" should not be based on the integrated CC-BED domain of Pm6SI. We have removed these claims from the revised version to prevent confusion.

2) It is puzzling that after a detailed EMS mutagenesis study that clearly identifies the candidate gene, the authors choose to use an over-expression promoter (*ubi*) and the cDNA of the CNL candidate as their gene construct for the transgenic complementation. Over-expression constructs can mask pathogen specificity tests. Why was the native promoter and the full genomic sequence of the encoding region not used in the transgenic studies? The authors can at least provide this data and can be compared with the overexpressing Pm6SI-cDNA construct.

>Response: Thank you for your suggestion. Based on map-based cloning and validation with up to 30 independent EMS mutants and VIGS, we were already highly confident that the candidate gene *CNL1* was indeed *Pm6SI*. This conclusion is supported by the fact that at least 13 published wheat and barley R genes have been validated using mutagenesis

alone or in combination with VIGS (*Pm2*, *Lr22a*, *Rph1*, *Yr5*, *YrSP*, *Yr7*, *Sm1*, *SuSr-D1*, *Lr14a*, *Yr27*, *Snn3-D1*, *Lr9*, and *Pm69*).

We fully agree with you that, in some cases, overexpression of certain R genes may lead to general resistance. To determine whether the powdery mildew resistance observed in *Pm6SI* transgenic plants is attributable to *Pm6SI* overexpression, we investigated expression levels and resistance phenotypes of 24 transgenic T₁ families that include 15 in the previous version and nine new T₁ families. Notably, two *Pm6SI* transgenic T₁ families, L17 and L21, exhibited similar transgenic expression levels to recombinant T3012II-3 but maintained powdery mildew resistance like control T3012II-3 (Fig. 3). This indicates that the resistance observed in *Pm6SI* transgenic T₁ plants is unlikely due to *Pm6SI* overexpression.

In addition, according to your suggestion, we constructed a vector with the native promoter and the full genomic sequence of the *Pm6SI* coding region. Transgenic complementation experiments are currently underway.

3) What is missing from the study is to what extent *Pm6SI* confers resistance to multiple *Bgt* isolates as the authors make claims to its breeding value. I can only see data presented for a single *Bgt* isolate. This information is critical in defining the usefulness of *Pm6SI*. Previous studies about the spectrum of isolates on the chromosome harbouring *Pm6SI* was based on the full chromosome addition line. With the availability of the smallest introgression segment with *Pm6SI*, the authors should provide data using this genetic stock rather than relying on previous addition/full chromosome substitution line. The transgenic line using the over-expressing construct will be inappropriate. (The authors will be well aware of the previous *Pm21* over-expression with a candidate kinase that was later corrected when it became evident that *Pm21* encoded a CNL).

>Response: We fully agree with you that the resistance spectrum analysis of *Pm6SI* in our paper is important for assessing its breeding value, and conducting this analysis using over-expressed transgenic lines would be inappropriate. Following your advice, we conducted a resistance assessment of *CS-Ae. longissima* recombinant line T3012II-3 containing small segments harboring *Pm6SI*, using 36 currently available *Bgt* isolates. The results showed that 35 of the 36 *Bgt* isolates were avirulent on both T3012II-3 and the resistant control TA7548. No difference in resistance spectrum was observed between recombinant line T3012II-3 and 6S^I addition line TA7548. These results are detailed in Supplementary Table 7.

We agree with you that the transgenic line using the over-expressing construct will be inappropriate. We have engineered a construct containing the native promoter and the complete genomic sequence of the *Pm6SI* coding region. Transgenic complementation experiments are presently ongoing.

4) Again, it is noted that the whole chromosome addition line (TA7548) was what was used in the cell death and intracellular reactive oxygen species studies shown in Figure 3 and

described in the Results. The authors should provide TPN-Coomassie blue and DAB-Coomassie blue data using the small introgressed segment harbouring Pm6Sl for any claims attributed to the Pm6Sl gene. Similarly with Figure 2, the powdery mildew resistance phenotype should show what can be unequivocally attributed to Pm6Sl by using recombinant T3012II-3 containing Pm6Sl in the smallest 6Sl segment, rather than TA7548. After all the emphasis of the manuscript is about the isolation of the Pm6Sl gene not the original chromosome addition (TA7548) into wheat which has previously been published.

>Response: Following your suggestion, we re-investigated cell death and intracellular reactive oxygen species using the smallest introgressed segment carrying *Pm6Sl* (T3012II-3) and revised the results in Fig. 4c and 4d. In addition, we replaced the resistant control TA7548 in Fig. 3 with T3012II-3 to more definitively attribute the observed effects to Pm6Sl. For the VIGS experiment in Fig. 2b, we retained the 6S¹ addition line TA7548, as it more effectively demonstrates that knockdown of *Pm6Sl* gene on chromosome 6S¹ can convert disease resistance to susceptibility.

Other comments:

The section of the manuscript entitled “Pm6Sl is only present in *Aegilops longissima*” is rather premature. Currently the database of wheat relatives from the Sitopsis section is limited and hence any claims to Pm6Sl being confined to *Ae longissimi* should await more genome sequences for diverse accessions of the Sitopsis (S genome) species. Consequently, the heading in line 246 and statements in lines 251-252 needs to be revised.

>Response: We agree with you and revised the heading (see Line 276, ‘Pm6Sl is likely present exclusively in *Aegilops longissima*’) and the statement (see Lines 282-283, ‘Therefore, it is assumed that Pm6Sl may be present exclusively in *Ae. longissima*, although further research is necessary to substantiate this assumption.’).

The manuscript will benefit from an English revision.

>Response: We have improved our manuscript through English Language Editing by SPRINGER NATURE.

Some glaring areas include:

Line 57 insert ...Pm69 ‘which’ encode

>Response: Thank you for pointing out the error, the sentence has been updated (see Line 57).

Line 66-67 rephrase the sentences

>Response: We have rewritten this sentence (see Lines 66-67).

Line 98 replace temperately with temporarily

>Response: We have deleted it (see Line 99).

Line 111 replace presents with represents

>Response: The sentence has been updated (see Lines 110-114).

Line 129-130 Rephrase the sentence

>Response: We have rephrased this sentence (see Lines 131-133). ‘By integrating the *Bgt* responses of the eight recombinants via five-marker analysis, we further delimited *Pm6Sl* to a 210 kb region (654.21-654.42 Mb) between *Ae/39960* and *Ae/09126*.’

Line 376 insert were between leaves and fully

>Response: We have inserted it (see Line 437).

Line 420-421, remove were from seedling and segregated

>Response: We have removed it (see Line 480).

Reviewers' Comments:

Reviewer #2:

Remarks to the Author:

I have carefully read over the author rebuttals to the three reviews. In my opinion, they have addressed all of them in a satisfactory manner. I think the manuscript in its revised form will be acceptable for publications.

Reviewer #3:

Remarks to the Author:

The authors have addressed the key questions raised from my previous review and has improved the findings of the overall study. Well done.

The use of the smaller introgressed segment, T3012II-3 rather than the whole chromosome addition, TA7548, adds credibility to the broad range of resistance to Bgt isolates attributed to Pm6Sl.

In keeping with the caution from making inferences from the limited pan-genome of Triticeae species, reiterated in my earlier review, its inappropriate to use the subheading “Pm6Sl is likely present exclusively in *Aegilops longissima*” in line 276. The authors have correctly provided guarded comments about this observation in the context of the section. A more appropriate title for the section is “Pm6Sl relationships with NLRs from Gramineae species”.

The authors indicate that they have made progress with wheat transgenics carrying the native promoter and genomic sequence of Pm6Sl. Why don't the authors include the information in this paper as there are no benefits from it being published elsewhere. They have already shown that Pm6Sl confers all stage resistance, and so can at least show that in T0 plants the use of the native promoter and genomic clone also confers resistance.

The authors should correct the revised statement at the end of the Introduction section. The claim made in the sentence, “This study documents a resistance gene for elucidating the molecular mechanisms underlying gene-for-gene interactions between wheat and its pathogens” is an exaggeration. There are many different wheat pathogens, and this study only deals with the Bgt pathogen.

REVIEWER COMMENTS

Reviewer #2 (Remarks to the Author):

I have carefully read over the author rebuttals to the three reviews. In my opinion, they have addressed all of them in a satisfactory manner. I think the manuscript in its revised form will be acceptable for publications.

We thank you for your encouraging comments.

Reviewer #3 (Remarks to the Author):

The authors have addressed the key questions raised from my previous review and has improved the findings of the overall study. Well done.

The use of the smaller introgressed segment, T3012II-3 rather than the whole chromosome addition, TA7548, adds credibility to the broad range of resistance to Bgt isolates attributed to Pm6SI.

We thank you for your valuable comments, which have helped to improve the quality and rigor of our manuscript.

In keeping with the caution from making inferences from the limited pan-genome of Triticeae species, reiterated in my earlier review, its inappropriate to use the subheading “Pm6SI is likely present exclusively in *Aegilops longissima*” in line 276. The authors have correctly provided guarded comments about this observation in the context of the section. A more appropriate title for the section is “Pm6SI relationships with NLRs from Gramineae species”.

>Response: Thank you for your suggestion. The subheading in line 284 has been modified as you suggested.

The authors indicate that they have made progress with wheat transgenics carrying the native promoter and genomic sequence of Pm6SI. Why don't the authors include the information in this paper as there are no benefits from it being published elsewhere. They have already shown that Pm6SI confers all stage resistance, and so can at least show that in T₀ plants the use of the native promoter and genomic clone also confers resistance.

>Response: Your commendable recommendation has been incorporated into our subsequent experiments. Using *CNL1*'s native promoter and genomic sequence, we successfully established a vector spanning 9,727 bp, which includes a presumed 4,569 bp native promoter, 4,496 bp exon and intron sequences, and a 662 bp terminator of *CNL1*. This vector was used for wheat genetic transformation, yielding six T₀-generation transgenic wheat seedlings on August 6, 2024.

To confirm *CNL1*'s native promoter-driven function, we selected a completely expanded new leaf from each T₀ seedling, one half for expression level analysis and the other half for powdery mildew resistance assessment. As the results, all T₀ seedlings tested positive for *CNL1* and showed powdery mildew resistance comparable to resistance

control T3012II-3. However, each seedling displayed varying levels of *CNL1* expression compared to control T3012II-3. Thus, our transgenic assay using the *CNL1* genomic sequence under its native promoter further supports our earlier assertion that *CNL1*, identified as the *Pm6Sl* gene, confers resistance in wheat.

These important findings were incorporated into the manuscript (lines 199-204), enhancing the overall significance and rigor of our study.

The authors should correct the revised statement at the end of the Introduction section. The claim made in the sentence, "This study documents a resistance gene for elucidating the molecular mechanisms underlying gene-for-gene interactions between wheat and its pathogens" is an exaggeration. There are many different wheat pathogens, and this study only deals with the *Bgt* pathogen.

>Response: Thank you for reviewing our manuscript critically and providing constructive feedback. We concur, and have revised "its pathogens" to "the *Bgt* pathogen" (See line 112).

Reviewers' Comments:

Reviewer #3:

Remarks to the Author:

The authors have adequately incorporated the recommendations made in my last review. There are no further technical items that need to be addressed.

Well done to the team.

REVIEWER COMMENTS

Reviewer #3 (Remarks to the Author):

The authors have adequately incorporated the recommendations made in my last review.

There are no further technical items that need to be addressed.

Well done to the team.

We thank you for your encouraging comments.